# CHARTMASTER: ADVANCING CHART-TO-CODE GENERATION WITH REAL-WORLD CHARTS AND CHART SIMILARITY REINFORCEMENT LEARNING

## ABSTRACT

The chart-to-code generation task requires MLLMs to convert chart images into executable code. This task faces two main challenges: limited data diversity and the difficulty of maintaining visual consistency between generated charts and the original ones. Existing datasets mainly rely on synthetic seed data to prompt GPT models for code generation, resulting in homogeneous samples that limit model generalization to real-world chart styles. To address this, we propose **ReChart-Prompt**, leveraging real-world, human-designed charts extracted from arXiv papers as prompts. By harnessing the rich content and diverse visual styles of arXiv charts, we construct ReChartPrompt-240K, a large-scale and highly diverse dataset that better reflects realistic chart variations. For the second challenge, although SFT improves code understanding by optimizing next-token prediction, it does not provide direct supervision on visual features. As a result, it often fails to guarantee that the generated charts visually match the original ones. To address this, we propose **ChartSimRL**, a GRPO-based reinforcement learning algorithm guided by a novel chart similarity reward. This reward consists of two components: *attribute similarity*, which measures the overlap of chart attributes like layout and color between the generated and original charts, and *visual similarity*, which evaluates overall visual features, including texture, using convolutional neural networks. Unlike traditional text-based rewards, our reward accounts for the multimodal nature of the chart-to-code generation task, significantly enhancing the model's ability to accurately reproduce charts. Integrating ReChartPrompt and ChartSimRL, we develop the **ChartMaster** model, achieving SOTA results among 7B-parameter models and rivaling GPT-4o on various chart-to-code benchmarks. We will release all code, datasets, and models to facilitate further research.

## 1 INTRODUCTION

The chart-to-code generation task aims to automatically convert chart images into executable code (Yang et al., 2024a), enabling applications including automated data analysis, report generation, and intelligent question answering (Zhao et al., 2025; Rodriguez et al., 2024; Xia et al., 2023; Awal et al., 2025). This task is challenging as it requires accurate visual understanding, cross-modal reasoning, and advanced code synthesis. Although recent advances in Multimodal Large Language Models (MLLMs) show promising results in various vision-language tasks, their performance on chart-to-code generation remains limited due to the unique complexity of charts and the need for precise code output.

Prior work, such as ChartCoder (Zhao et al., 2025), advanced the field by building the large Chart2Code-160K dataset. This dataset is synthesized by guiding GPT-4o (Hurst et al., 2024) with predefined chart attributes like chart type, color, and text. While this approach reduces the need for costly manual annotations and achieves strong performance, relying on predefined attribute seeds can introduce homogeneity and limit variability in the resulting dataset (see Appendix Fig. 5), potentially restricting model generalization to diverse real-world charts.

To address this limitation, we introduce Real-world Chart Prompt Code Generation (ReChart-Prompt), a novel automated pipeline that extracts real chart images from arXiv papers and lever-

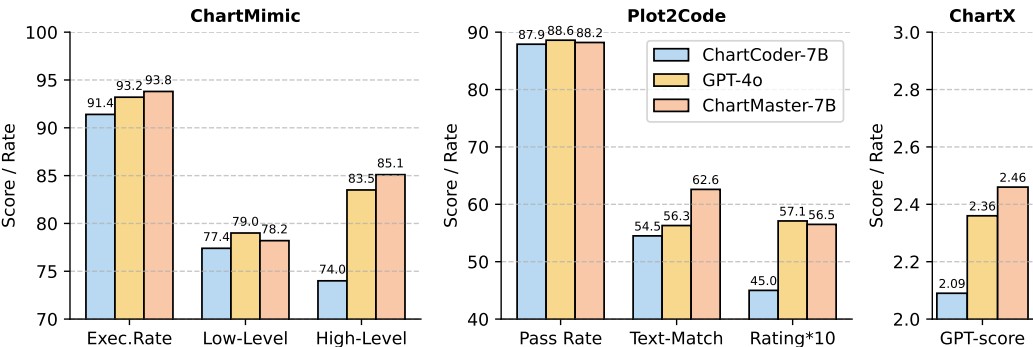

Figure 1: Performance comparison on three benchmarks. Our method outperforms ChartCoder-7B (Zhao et al., 2025), and matches or exceeds GPT-4o on certain metrics. For better representation, the "Rating" metric in the Plot2Code benchmark is multiplied by 10.

ages the Qwen2.5-VL-72B model (Bai et al., 2025) to generate corresponding code. By collecting 30,071 papers and utilizing their author-designed charts as prompts, we construct ReChartPrompt-240K, a large-scale dataset comprising 240K chart–code pairs. Since these charts originate from papers across diverse research fields and exhibit a wide variety of design styles, the dataset captures rich visual and semantic diversity, as illustrated in Fig. 5. This heterogeneity enables effective generalization to real-world scenarios.

While supervised fine-tuning with diverse data can help models generate better chart code, such next-token prediction alone does not ensure the output charts are visually faithful to the references. As shown in Fig. 4, the SFT model produces charts closer to the ground truth than the baseline, but noticeable discrepancies remain in color, element positioning, and other visual attributes. To address this, we propose ChartSimRL, a reinforcement learning algorithm based on Group Relative Policy Optimization (GRPO) (Shao et al., 2024), guided by a novel chart similarity reward. Specifically, the reward jointly considers (1) *attribute similarity* that evaluates the consistency of chart elements such as textual content, numerical values, layout and color, and (2) *visual similarity*, which assesses holistic visual resemblance using convolutional neural networks (e.g., ResNet (He et al., 2016)) to extract and compare visual features. To the best of our knowledge, this is the first reward system that explicitly enforces multimodal visual-semantic consistency for chart-to-code generation. By encouraging models to produce code that renders charts both semantically accurate and visually faithful, we address a critical gap in prior research and support more robust generalization to real-world chart reproduction.

In summary, we introduce ChartMaster, an efficient framework for chart-to-code generation that combines the ReChartPrompt data generation pipeline with the ChartSimRL reinforcement learning strategy. Our key contributions are: (1) ReChartPrompt, an automated method for generating diverse datasets from real-world charts; (2) ChartSimRL, a reinforcement learning algorithm that uses both visual and attribute similarity to improve output; and (3) ChartMaster-7B, a compact model that delivers near GPT-4o performance with only 7 billion parameters. Fig. 1 highlights its efficiency and effectiveness.

## 2 RELATED WORK

### 2.1 MULTIMODAL CODE GENERATION

Multimodal large language models (MLLMs) have recently demonstrated strong capabilities in code generation (Zhang et al., 2024a). Notably, MMCode (Li et al., 2024b) targets algorithmic problems embedded in visually rich contexts, where tasks are accompanied by one or more images.

Among multimodal code generation tasks, chart-to-code translation has emerged as a critical challenge (Yang et al., 2024b). Existing benchmarks include Design2Code (Si et al., 2024), which evaluates HTML generation using CLIP scores (Radford et al., 2021) and structural HTML metrics, and Plot2Code (Wu et al., 2024), which assesses both code correctness and visual fidelity. However, since the datasets for Design2Code and Plot2Code are sourced from the web, there is a risk of data

leakage, which may compromise the reliability of model evaluation. To address this issue, Chart-Mimic (Yang et al., 2024a) provides a manually curated dataset of 4,800 chart-code pairs, along with additional fine-grained evaluation metrics.

Despite these benchmarks, large-scale chart-to-code training datasets remain scarce. ChartCoder (Zhao et al., 2025) addresses this by creating Chart2Code-160K, the first large-scale training set generated by guiding GPT-4o with predefined chart attributes such as type, color, values, and titles. It further employs the "Snippet of Thought" strategy (Zheng et al., 2023; Luo et al., 2024) to decompose code generation into structured steps, significantly boosting chart reasoning. Yet, reliance on fixed attributes limits chart diversity. In contrast, our ReChartPrompt leverages real-world charts from arXiv papers as prompts, yielding more diverse and representative chart–code pairs.

## 2.2 REINFORCEMENT LEARNING FOR MLLMS

Reinforcement learning (RL) effectively enhances model capabilities (Wang et al., 2024b; Milani et al., 2024). For example, RL from human feedback (RLHF) (Bai et al., 2022) and direct preference optimization (DPO) (Rafailov et al., 2023) aligned model outputs with human preferences, improving complex reasoning and output quality. Building on these advances, Group Relative Policy Optimization (GRPO) (Shao et al., 2024) was proposed as a novel RL algorithm that updated policies using relative rewards computed from groups of samples. DeepSeek-R1 (Guo et al., 2025) employed simple yet effective rewards based on output accuracy and response format, which enabled stable training and emergent reasoning such as reflection and "a-ha" moments.

Inspired by DeepSeek-R1's success, recent work extended GRPO-based RL to MLLMs (Tan et al., 2025; Zhang et al., 2025b; Peng et al., 2025; Shen et al., 2025) in two main directions. The first adapts R1's method to MLLMs—for instance, Vision-R1 (Huang et al., 2025) uses SFT data with reflection for cold-start training and applies GRPO with accuracy- and format-based rewards. Similarly, MM-EUREKA (Meng et al., 2025) refines reward design and loss functions, successfully reproducing the visual "aha moment," where the model revisits images "upon closer inspection." These works primarily focus on mathematical reasoning tasks. The second direction applies GRPO to broader tasks such as chart understanding (Masry et al., 2025b), visual perception (Yu et al., 2025), segmentation (Liu et al., 2025), and grounding (Zhang et al., 2025a), demonstrating its robustness and generalizability across domains.

However, to our knowledge, GRPO has not been applied to chart-to-code generation, mainly due to the challenge of designing reward functions that encourage generated code to faithfully reproduce charts both semantically and visually. We address this by proposing a novel chart similarity reward, significantly improving chart reproduction quality.

## 3 METHOD

Fig. 2 illustrates the overall framework of ChartMaster, which consists of two main stages: data generation and model training.

### 3.1 USING REAL-WORLD CHARTS TO GENERATE DATASET

To improve dataset diversity, we use real-world chart images as input to guide code generation, as shown in Fig. 2 (a). This approach captures richer styles and content that predefined attribute seeds cannot represent.

**(1) Collecting Images from arXiv.** We leverage the arXiv API and Python's `requests` library to download paper source files, including LaTeX sources and image files (`.pdf`, `.png`, `.jpg`). To ensure diversity, we query source files related to top conferences (e.g., ICLR) and journals (e.g., TPAMI), extracting all images for subsequent processing.

**(2) Filtering Non-Chart Images.** Since extracted images include various diagrams beyond charts, we use the Qwen2.5-VL-72B model to classify images into 12 predefined chart categories. Images outside these categories are discarded. Classification is performed by prompting the model with $P_{type}$ (see Fig. 6 in Appendix) to assign chart types.

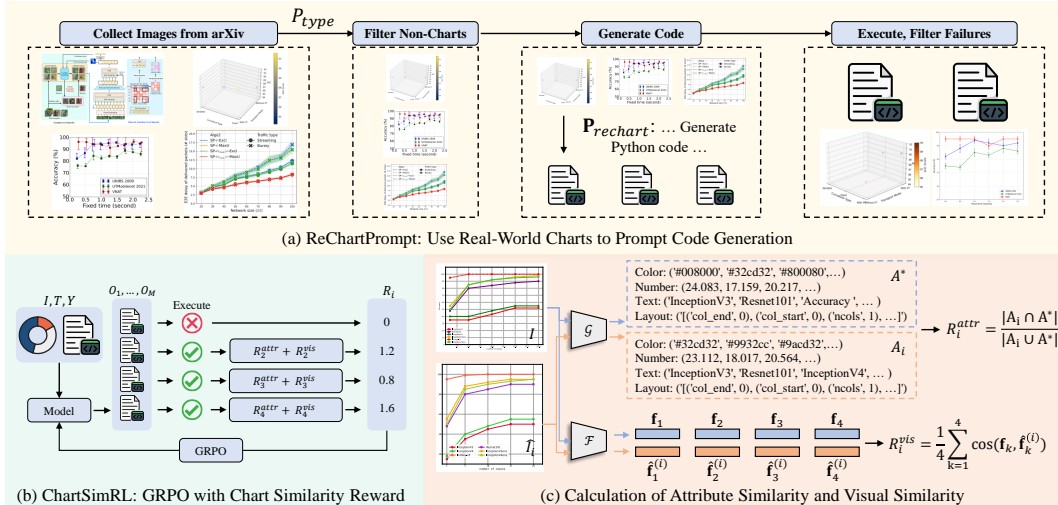

Figure 2: The overall framework of ChartMaster. (a) Real-world charts from arXiv are curated to create the ReChartPrompt-240K dataset for SFT (SFT is omitted in the figure). (b) The model is further optimized with ChartSimRL. (c) The definition of Chart Similarity: $\mathcal{G}$ denotes the semantic attribute extraction tool; $\mathcal{F}$ is the CNN-based feature extractor; and $\mathbf{f}$ is the extracted feature vector.

**(3) Generating code with ReChartPrompt.** The Qwen2.5-VL-72B model has demonstrated strong chart-to-code generation capabilities. As an open-source model, it is easily deployed via the vLLM framework (Kwon et al., 2023), making it well-suited for large-scale data generation. We design a set of 20 chart-to-code prompts to enrich instruction diversity, collectively referred to as $\mathbf{P}_{\text{rechart}}$ (see Fig. 7 in Appendix). Below is an example: *<Real-World Chart>Please generate Python matplotlib code to recreate the picture shown.*

**(4) Code Execution, Filtering, and Dataset Construction.** Generated code snippets may suffer from two issues: (a) execution errors caused by non-existent packages or syntax mistakes, and (b) discrepancies between the generated charts and the original images. To mitigate these problems, we execute all generated code and discard those that fail at runtime. We then pair the successfully executed code outputs with their generated images and instructions to form the final training triplets.

**Summary.** We download 30,071 papers from arXiv and extract their figures, filtering out non-chart ones to obtain 288,992 chart images. Using these charts, the Qwen2.5-VL-72B model generates corresponding code. After executing the generated code and removing failed cases, we collect 242,479 high-quality triplets that constitute the **ReChartPrompt-240K** dataset. Formally, the dataset is defined as $\mathcal{D} = \{(I_i, T_i, Y_i)\}_{i=1}^N$, where $I_i$ represents a chart image, $T_i \in \mathbf{P}_{\text{rechart}}$ is the instruction prompt, and $Y_i$ denotes the executable code. Notably, all real-world chart data and generation models employed in this process are open-source, ensuring minimal cost and excellent scalability.

## 3.2 TRAINING CHARTMASTER: SFT AND CHARTSIMRL

ChartMaster is trained in two stages: (1) SFT on the ReChartPrompt-240K dataset to establish a solid foundation; and (2) further optimized with ChartSimRL to address the limitations of SFT's next-token prediction in maintaining visual consistency.

**Supervised Fine-Tuning.** We conduct SFT by maximizing the likelihood of ground-truth code $Y_i$ given chart image $I_i$ and instruction $T_i$:

$$J_{\text{SFT}}(\theta) = -\frac{1}{N} \sum_{i=1}^{N} \log \pi_\theta(Y_i \mid I_i, T_i).$$

**Reinforcement Learning with ChartSimRL.** While SFT strengthens the model's basic capability, discrepancies may still exist between the generated charts and the originals (see Fig. 4). To further improve reproduction fidelity, we continue training the model using ChartSimRL, as illustrated in Fig. 2 (b). Specifically, for each training sample $(I, T, Y)$, the model samples a group of $M$

candidate codes:

$$\{O_1, O_2, \ldots, O_M\} \sim \pi_\theta(\cdot \mid I, T).$$

Each candidate code $O_i$ is then executed to generate a chart image $\hat{I}_i$, which is subsequently compared with the original chart $I_i$ to compute a Chart Similarity Reward. If the execution of $O_i$ fails, the corresponding reward is set to zero.

**Chart Similarity Reward.** Traditional reward functions, such as the accuracy reward used in (Guo et al., 2025; Huang et al., 2025), primarily assess the consistency between generated text and ground-truth text. However, the chart-to-code task is inherently multimodal, involving both code and generated charts, requiring evaluation of not only semantic correctness but also visual alignment. To this end, we design a novel chart similarity reward as:

$$R_i = R_i^{\text{attr}} + R_i^{\text{vis}}.$$

Here, $R_i^{\text{attr}}$ measures the semantic consistency, and $R_i^{\text{vis}}$ captures visual similarity (see Fig. 2 (c)).

*Attribute Similarity:* We develop a semantic attribute extraction tool based on the ChartMimic codebase Yang et al. (2024a), denoted $\mathcal{G}(\cdot)$, to obtain attribute sets from chart images and their code. Given $\mathcal{A}_i = \mathcal{G}(\hat{I}_i, O_i)$ and $\mathcal{A}^* = \mathcal{G}(I, Y)$, the semantic similarity $R_i^{\text{attr}}$ is computed as their Jaccard similarity:

$$R_i^{\text{attr}} = \frac{|\mathcal{A}_i \cap \mathcal{A}^*|}{|\mathcal{A}_i \cup \mathcal{A}^*|} \in [0, 1].$$

By design, $R_i^{\text{attr}} = 1$ indicates a perfect match of semantic attributes, while lower values reflect semantic discrepancies. To accommodate minor numerical variations, we consider numerical values $a \in \mathcal{A}_i$ and $b \in \mathcal{A}^*$ matching if $|a - b| \leq 0.01 \times |b|$.

*Visual Similarity:* We use a pretrained ResNet-18 network (He et al., 2016) $\mathcal{F} = \{\mathcal{F}_1, \mathcal{F}_2, \mathcal{F}_3, \mathcal{F}_4\}$ to extract feature maps from both $I$ and $\hat{I}_i$. Here, $\mathcal{F}_k(\cdot) \in \mathbb{R}^{C_k \times H_k \times W_k}$ denotes the output feature map of the $k$-th residual block. We extract the feature map and flatten them into vectors like:

$$F_k = \mathcal{F}_k(I), \qquad \hat{F}_k^{(i)} = \mathcal{F}_k(\hat{I}_i),$$
$$\mathbf{f}_k = \text{vec}(F_k) \in \mathbb{R}^{d_k}, \qquad \hat{\mathbf{f}}_k^{(i)} = \text{vec}(\hat{F}_k^{(i)}) \in \mathbb{R}^{d_k},$$

where $d_k = C_k \times H_k \times W_k$. The visual similarity reward is defined as the average cosine similarity between the corresponding feature vectors,

$$R_i^{\text{vis}} = \frac{1}{4} \sum_{k=1}^{4} \frac{\mathbf{f}_k \cdot \hat{\mathbf{f}}_k^{(i)}}{\|\mathbf{f}_k\| \, \|\hat{\mathbf{f}}_k^{(i)}\|} \in [0, 1].$$

**Chart Similarity Reinforcement Learning.** We normalize rewards within a group of $M$ candidates to compute relative advantages:

$$\hat{A}_i = \frac{R_i - \text{mean}(\{R_j\}_{j=1}^M)}{\text{std}(\{R_j\}_{j=1}^M)},$$

where $\text{mean}(\cdot)$ and $\text{std}(\cdot)$ denote the sample mean and standard deviation, respectively.

Following the GRPO framework (Shao et al., 2024), we update the model by maximizing the clipped surrogate objective with a KL penalty to stabilize training:

$$J_{\text{ChartSimRL}}(\theta) = \mathbb{E}_{(I,T) \sim p_\mathcal{D}, \, \{o_i\}_{i=1}^M \sim \pi_{\text{old}}(\cdot|I,T)} \left[ \frac{1}{M} \sum_{i=1}^{M} \min \left( \frac{\pi_\theta(o_i|I,T)}{\pi_{\text{old}}(o_i|I,T)} \hat{A}_i, \right. \right.$$

$$\left. \left. \text{clip}\left( \frac{\pi_\theta(o_i|I,T)}{\pi_{\text{old}}(o_i|I,T)}, 1 - \epsilon, 1 + \epsilon \right) \hat{A}_i \right) - \beta D_{\text{KL}}\big(\pi_\theta(\cdot|I,T) \| \pi_{\text{ref}}(\cdot|I,T)\big) \right],$$

where $\pi_{\text{old}}$ is the previous policy, $\pi_{\text{ref}}$ is the reference policy, $\epsilon$ is the clipping hyperparameter, and $\beta$ controls the KL regularization strength.

ChartSimRL guides the model to generate chart code that better aligns with the original charts' semantic and visual properties, significantly improving chart-to-code generation performance beyond what is achievable by supervised fine-tuning alone.

Table 1: Evaluation results of various MLLMs. Reported results are taken from existing benchmarks when available; missing results are supplemented using official codebases and are marked with *. Among open-source 7B-scale models, our method achieves the best performance.

| Model | ChartMimic | | | Plot2Code | | | ChartX |
|---|---|---|---|---|---|---|---|
| | Exec.Rate | Low-Level | High-Level | Pass Rate | Text-Match | Rating | GPT-score |
| Full score | 100 | 100 | 100 | 100 | 100 | 10 | 5 |
| *Closed-Source Model* | | | | | | | |
| GeminiProVision (Team et al., 2023) | 68.2 | 53.8 | 53.3 | 68.2 | 53.6 | 3.69 | - |
| Claude-3-opus (Anthropic, 2024) | 83.3 | 60.5 | 60.1 | 84.1 | 57.5 | 3.80 | - |
| GPT-4V (Hurst et al., 2024) | 91.2 | 76.4 | 78.9 | 84.1 | 57.7 | 5.58 | 2.63 |
| GPT-4o (Hurst et al., 2024) | 93.2 | 79.0 | 83.5 | 88.6 | 56.3 | 5.71 | 2.36* |
| *Open-Source Model* | | | | | | | |
| ChartAssisstant-13B (Meng et al., 2024) | - | - | - | - | - | - | 0.82 |
| ChartVLM-L-14B (Xia et al., 2024) | 19.5 | 15.8 | 13.9 | - | - | - | 1.58 |
| DeepSeek-VL-7B (Lu et al., 2024) | 41.3 | 19.0 | 17.6 | 64.4 | 32.6 | 2.26 | - |
| TinyChart-3B (Zhang et al., 2024b) | 42.5 | 26.3 | 25.9 | 43.2 | 44.6 | 2.19 | 1.89 |
| ChartLlama-13B (Han et al., 2023) | 57.5 | 24.8 | 28.1 | 58.4 | 40.3 | 2.32 | 0.94 |
| LLaVA-Next-Mistral-7B (Li et al., 2024a) | 59.7 | 20.7 | 21.3 | 72.0 | 38.7 | 2.87 | - |
| InternVL2-8B (Chen et al., 2024) | 61.8 | 34.4 | 38.9 | 77.3 | 37.1 | 2.78 | 1.63 |
| Qwen2-VL-7B (Wang et al., 2024a) | 67.0 | 32.9 | 35.0 | 68.2 | 33.8 | 3.10 | 1.50 |
| MiniCPM-Llama3-V2.5-8B (Yao et al., 2024) | 80.3 | 36.6 | 42.1 | 76.3 | 37.3 | 2.61 | 1.66 |
| Qwen2-VL-72B (Wang et al., 2024a) | 73.3 | 54.4 | 50.9 | 72.0 | 53.4 | 4.26 | 1.69 |
| InternVL2-Llama3-76B (Chen et al., 2024) | 83.2 | 54.8 | 62.2 | 85.6 | 46.6 | 3.89 | 1.74 |
| Qwen2.5-VL-72B* (Bai et al., 2025) | 88.5 | 72.7 | 79.1 | 84.8 | **68.4** | **6.83** | **2.52** |
| ChartCoder-7B (Zhao et al., 2025) | 91.4 | 77.4 | 74.0 | 87.9 | 54.5 | 4.50 | 2.09 |
| Qwen2.5-VL-7B* (Baseline) (Bai et al., 2025) | 65.5 | 39.9 | 40.7 | 67.4 | 43.8 | 4.60 | 2.18 |
| ChartMaster-7B | **93.8** | **78.2** | **85.1** | **88.2** | 62.6 | 5.65 | 2.46 |

**Summary.** ReChartPrompt and ChartSimRL have been effectively integrated into the ChartMaster framework. This framework not only leverages real-world data for enhanced data diversity but also employs a novel algorithm to ensure visual and semantic alignment in chart reproduction. Consequently, ChartMaster stands as a comprehensive solution for the chart-to-code generation task, demonstrating marked improvements in performance and generalization capabilities.

## 4 EXPERIMENT

### 4.1 COMPARISON WITH SOTA

We instantiate ChartMaster on the Qwen2.5-VL-7B backbone, resulting in the ChartMaster-7B model, and conduct comprehensive comparisons with a range of MLLMs. The detailed implementation and evaluation protocols are provided in the Appendix B. As shown in Table 1, ChartMaster-7B achieves state-of-the-art performance among open-source models at the 7B scale, showing competitive performance against GPT-4o. Notably, ChartMaster-7B consistently outperforms the baseline Qwen2.5-VL-7B across all metrics; for instance, in the ChartMimic benchmark, it improves both low-level and high-level metrics by about 40 percentage points. Furthermore, although our training dataset is derived from the larger Qwen2.5-VL-72B model—essentially a distillation-like setting—ChartMaster-7B still surpasses Qwen2.5-VL-72B on several benchmarks. These results convincingly demonstrate the effectiveness of the ChartMaster framework.

### 4.2 ABLATION STUDY

**Ablation study on ChartMaster.** To assess the contribution of each component, we conduct an ablation study as summarized in Table 2. The base Qwen2.5-VL-7B model, without ReChartPrompt or ChartSimRL, demonstrates limited performance across benchmarks, revealing its restricted ability in both code generation and visual/semantic understanding. SFT with the ReChartPrompt-240K dataset leads to significant improvements in all metrics, demonstrating the high quality and effectiveness of ReChartPrompt-240K for chart-to-code generation. Additionally, applying ChartSimRL alone also significantly improves the baseline model's performance. This enhancement is attributed to our well-designed reward function, which effectively captures the semantic and visual features of the charts, optimizing the model's ability to generate code that closely aligns with the original

Table 2: Ablation study on the contribution of each key component.

| ReChartPrompt | ChartSimRL | ChartMimic | | | Plot2Code | | | ChartX |
|---|---|---|---|---|---|---|---|---|
| | | Exec.Rate | Low-Level | High-Level | Pass Rate | Text-Match | Rating | GPT-score |
| | | 65.5 | 39.9 | 40.7 | 67.4 | 43.8 | 4.60 | 2.18 |
| ✓ | | 91.1 | 73.7 | 80.9 | 80.3 | 59.3 | 5.34 | 2.36 |
| | ✓ | 83.6 | 58.6 | 57.6 | 72.7 | 50.8 | 5.19 | 2.23 |
| ✓ | ✓ | 93.8 | 78.2 | 85.1 | 88.2 | 62.6 | 5.65 | 2.46 |

Table 3: Ablation study of the Attribute and Visual similarity components in Chart-SimRL.

| $R_i^{\text{attr}}$ | $R_i^{\text{vis}}$ | ChartMimic | | |
|---|---|---|---|---|
| | | Exec.Rate | Low-Level | High-Level |
| | | 91.1 | 73.7 | 80.9 |
| ✓ | | 92.1 | 76.2 | 83.9 |
| | ✓ | 92.1 | 77.7 | 84.3 |
| ✓ | ✓ | 93.8 | 78.2 | 85.1 |

Table 4: Ablation study of different attribute similarity metrics on the ChartMimic benchmark.

| $R_i^{\text{attr}}$ | Formula | ChartMimic | | |
|---|---|---|---|---|
| | | Exec.Rate | Low-Level | High-Level |
| - | - | 91.1 | 73.7 | 80.9 |
| Precision | $\frac{|\mathcal{A}_i \cap \mathcal{A}^*|}{|\mathcal{A}_i|}$ | 90.0 | 72.6 | 79.0 |
| Recall | $\frac{|\mathcal{A}_i \cap \mathcal{A}^*|}{|\mathcal{A}^*|}$ | 90.6 | 74.7 | 81.7 |
| F1 | $\frac{|\mathcal{A}_i \cap \mathcal{A}^*|}{(|\mathcal{A}_i|+|\mathcal{A}^*|)/2}$ | 91.6 | 75.4 | **84.5** |
| Jaccard | $\frac{|\mathcal{A}_i \cap \mathcal{A}^*|}{|\mathcal{A}_i \cup \mathcal{A}^*|}$ | **92.1** | **76.2** | 83.9 |

charts. Therefore, further applying ChartSimRL on top of ReChartPrompt yields consistent performance gains, achieving optimal results.

**Ablation study on ChartSimRL.** ChartSimRL introduces a novel multimodal chart similarity reward that combines both semantic similarity ($R_i^{\text{attr}}$) and visual similarity ($R_i^{\text{vis}}$) between the candidate and original charts. To dissect the contribution of each component, we conduct ablation experiments summarized in Table 3. The results show that employing either $R_i^{\text{attr}}$ or $R_i^{\text{vis}}$ alone consistently improves performance across all evaluated metrics. Notably, the visual similarity reward yields more substantial gains, underscoring the critical importance of preserving visual fidelity in chart-to-code generation. Moreover, combining both rewards achieves the best overall results, demonstrating the advantage of a multi-faceted reward design that simultaneously captures semantic and visual aspects.

**Ablation study on Attribute Similarity.** We adopt Jaccard similarity as a stringent metric for attribute similarity, whereby a candidate table achieves a perfect score only if its attribute set exactly matches that of the ground truth; even minor discrepancies incur penalties. To thoroughly assess the impact of different attribute similarity measures—Precision, Recall, F1 score, and Jaccard similarity—we conduct experiments on the ChartMimic benchmark, with results summarized in Table 4.

Our findings indicate that optimizing exclusively for Precision may lead to a slight decline in overall performance, as the model can achieve high Precision by predicting a limited subset of correct attributes while neglecting overall coverage. In contrast, Recall emphasizes coverage, which helps mitigate this issue and yields modest improvements. The F1 score, by harmoniously balancing Precision and Recall, further alleviates extreme biases and delivers enhanced overall performance. Notably, Jaccard similarity, measuring the intersection over union between predicted and reference attribute sets, enforces stricter penalties on both missing and redundant attributes. This higher overlap requirement enables Jaccard similarity to more faithfully capture the true semantic similarity between attribute sets, thereby resulting in the best overall performance.

**Ablation Study on Visual Similarity.** We use ResNet-18 (He et al., 2016) to extract features from charts to compute visual similarity. In fact, there are numerous methods to measure the similarity between two charts. To investigate the impact of different visual similarity metrics on model performance, we conduct an ablation study summarized in Table 5. Standard metrics such as MSE, SSIM (Wang et al., 2004), and PSNR (Hore & Ziou, 2010) primarily evaluate pixel-level or structural similarity (details in Appendix C). The table shows that these metrics generally perform worse than more advanced methods. Notably, SSIM exhibits a significant decline in performance, indicating that pixel-based measures struggle to capture the complex visual nuances necessary for effective chart-to-code generation.

In contrast, CNN-based metrics like AlexNet (Krizhevsky et al., 2012), VGG (Simonyan & Zisserman, 2014), and ResNet (He et al., 2016), which compare features in a learned representation space, consistently outperform both the baseline and pixel-level metrics across all evaluation criteria. Among them, ResNet-18 achieves the highest performance, highlighting the effectiveness of deep visual features.

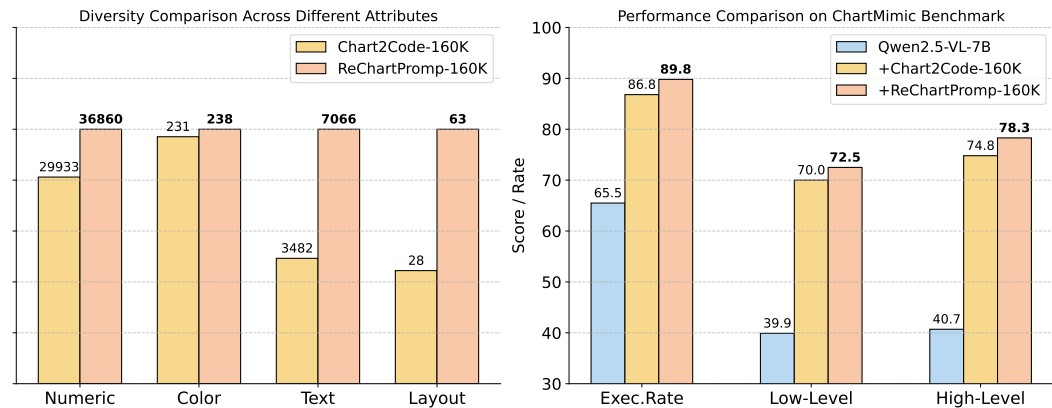

Figure 3: Comparison of diversity and fine-tuning results between Chart2Code-160K and ReChartPrompt-160K datasets.

For MLLM-based metrics, we leverage the high-level similarity prompt from ChartMimic combined with Qwen-2.5-VL-72B to evaluate the similarity between generated and reference charts. These metrics show improvements over the baseline. However, they still fall short of the best CNN-based metrics, suggesting that although MLLMs possess strong semantic understanding, further optimization is required for specialized visual tasks such as chart-to-code generation.

**Comparison with Advanced Dataset.** To comprehensively evaluate the diversity and quality of our dataset, we compare it with Chart2Code-160K (Zhao et al., 2025). For a fair comparison, we randomly sample 160K instances from our full dataset to construct the ReChartPrompt-160K subset. Using the attribute extraction tool $\mathcal{G}(\cdot)$, we count unique chart attributes—including numerical values, colors, textual elements, and layouts—in both datasets. A higher number of unique attributes indicates greater attribute diversity. As shown in the left panel of Fig. 3, ReChartPrompt-160K exhibits a substantially richer attribute distribution across all categories, notably in text and layout. This advantage stems primarily from

Table 5: Ablation study of different visual similarity metrics on the ChartMimic benchmark.

| $R_i^{\text{vis}}$ | ChartMimic | | |
| --- | --- | --- | --- |
| | Exec.Rate | Low-Level | High-Level |
| - | 91.1 | 73.7 | 80.9 |
| *Standard Metrics:* | | | |
| MSE | 91.1 | 73.6 | 77.9 |
| SSIM | 82.5 | 65.2 | 74.6 |
| PSNR | 91.4 | 75.1 | 82.1 |
| *CNN-Based Metrics:* | | | |
| AlexNet | 90.3 | 74.7 | 82.6 |
| VGG | 91.3 | 75.5 | 83.3 |
| ResNet-18 | **92.1** | **77.7** | **84.3** |
| *MLLM-Based Metrics:* | | | |
| Qwen-2.5-VL-72B | 91.7 | 77.5 | 83.9 |

Chart2Code-160K's reliance on seed data sources, which results in repeated attribute patterns, whereas ReChartPromp-160K samples from distinct arXiv papers, ensuring broader coverage and less redundancy (see Appendix Fig. 5). This higher diversity brings clear benefits: as illustrated in the right panel, models fine-tuned on ReChartPromp-160K consistently outperform those trained on Chart2Code-160K, demonstrating the importance of attribute diversity for robust and effective chart understanding and code generation.

### 4.3 QUALITATIVE ANALYSIS

Based on extensive experiments, we observe that ReChartPrompt generates charts with diverse and rich attributes, enabling the construction of a high-quality dataset that substantially enhances model performance. Building upon the distinctive features of the chart-to-code generation task, we propose the ChartSimRL algorithm, which further enhances the model's capabilities. To comprehensively analyze the improvements brought by these contributions, we conduct a qualitative comparison of generated charts at different training stages on the ChartMimic benchmark (Fig. 4). Our key findings are summarized as follows: **(1) The baseline model produces basic chart layouts but often fails to replicate fine-grained visual details,** leading to noticeable discrepancies between generated outputs and reference charts. **(2) Fine-tuning the base model on our ReChartPrompt-240K dataset ("Base.+ReCha.") significantly improves chart-to-code generation accuracy.** This improvement arises from the diverse, high-quality training data generated by conditioning on real-world

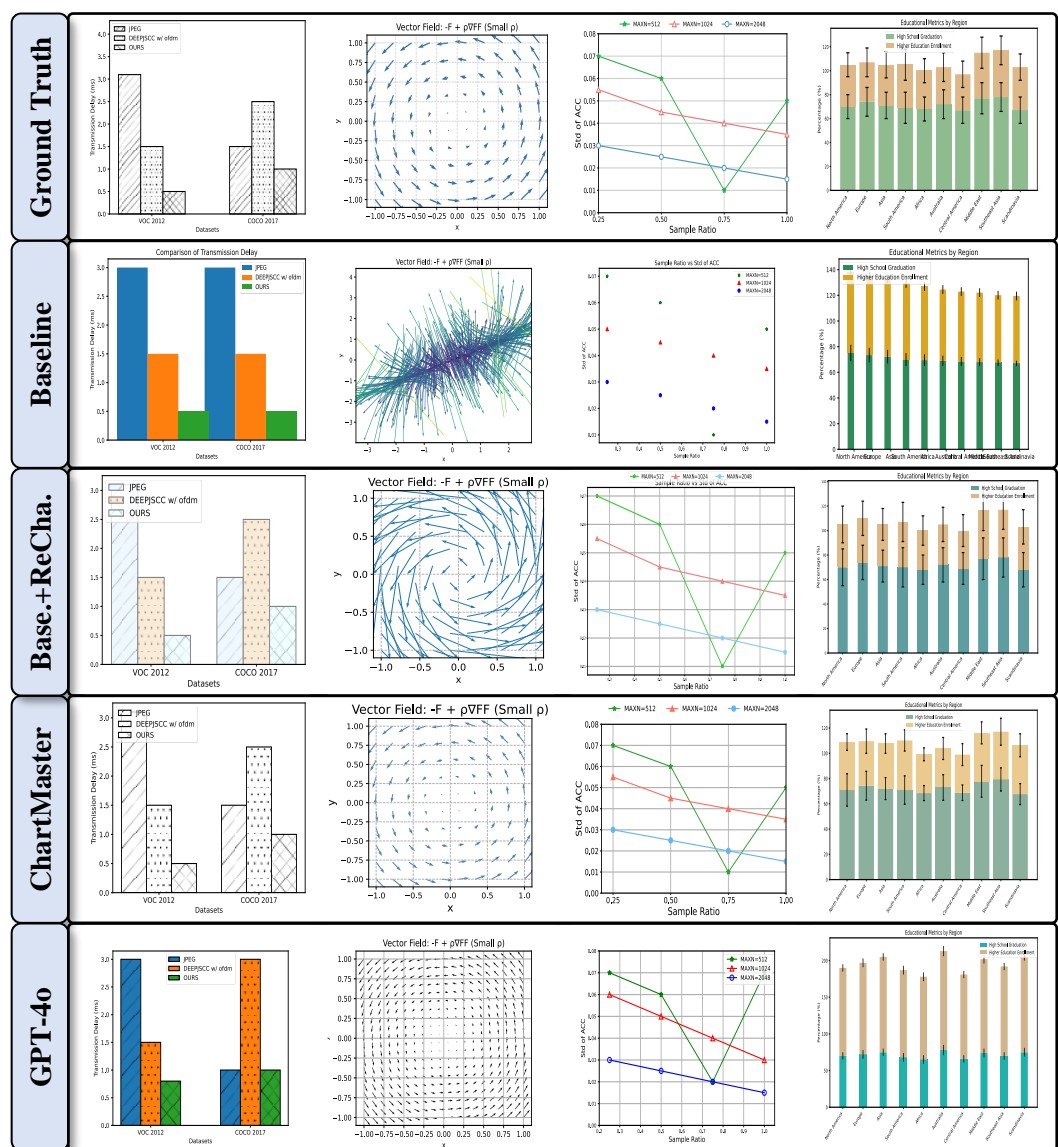

Figure 4: The test results of various models on the ChartMimic benchmark. "Base.+ReCha." refers to the baseline model fine-tuned with the ReChartPrompt-240K dataset. Incorporating ReChartPrompt significantly enhances the chart-to-code generation capability of the base model, while ChartSimRL further improves the handling of fine details.

chart prompts. Nonetheless, minor issues remain, such as slight mismatches in color or element positioning compared to the ground truth, indicating that supervised fine-tuning alone does not achieve perfect visual consistency. **(3) Incorporating the ChartSimRL algorithm further improves both visual and semantic alignment.** Notably, the model demonstrates enhanced color accuracy (as seen in the first column of Fig. 4) and more faithful reproduction of arrow styles in the second column, reflecting improved attention to key factual details. **(4) ChartMaster competes favorably with GPT-4o.** Notably, the ChartMaster-7B model can generate charts that more closely resemble the ground truth than those from GPT-4o, especially excelling in "mimicking" chart attributes. Additional generation results in Appendix Fig. 8 consistently support these conclusions.

## 5 CONCLUSION

In this paper, we propose **ChartMaster**, a novel chart-to-code generation framework paired with a tailored reinforcement learning algorithm. By introducing **ReChartPrompt**, we address data homogeneity issues in prior work and build a highly diverse ReChartPrompt-240K dataset. Our **ChartSimRL** algorithm combines semantic and visual similarity rewards, enabling the model to generate chart code that closely matches original visuals. Experiments show ChartMaster achieves performance on par with GPT-4o in chart-to-code tasks. We will open source all resources to foster community development and advance research in this area.

Beyond its technical innovations, ChartMaster supports automated scientific reporting and empowers data-driven decision-making across a wide range of domains. While our current framework targets common chart types and Python-based code, expanding its scope to include a wider range of chart formats and programming languages is an exciting direction for future work.

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

**Chart2Code-160K**

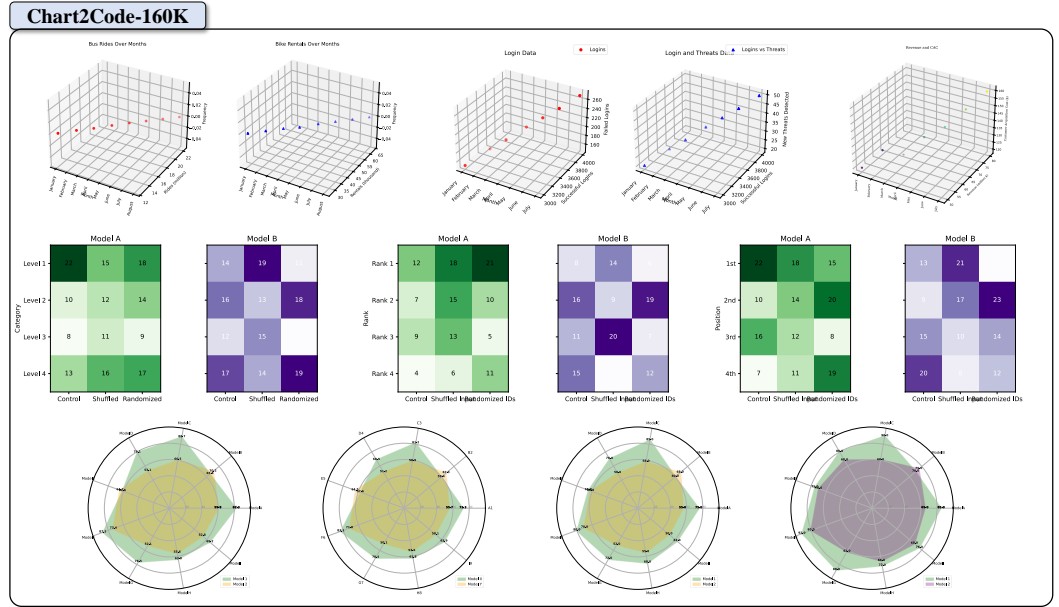

**ReChartPrompt**

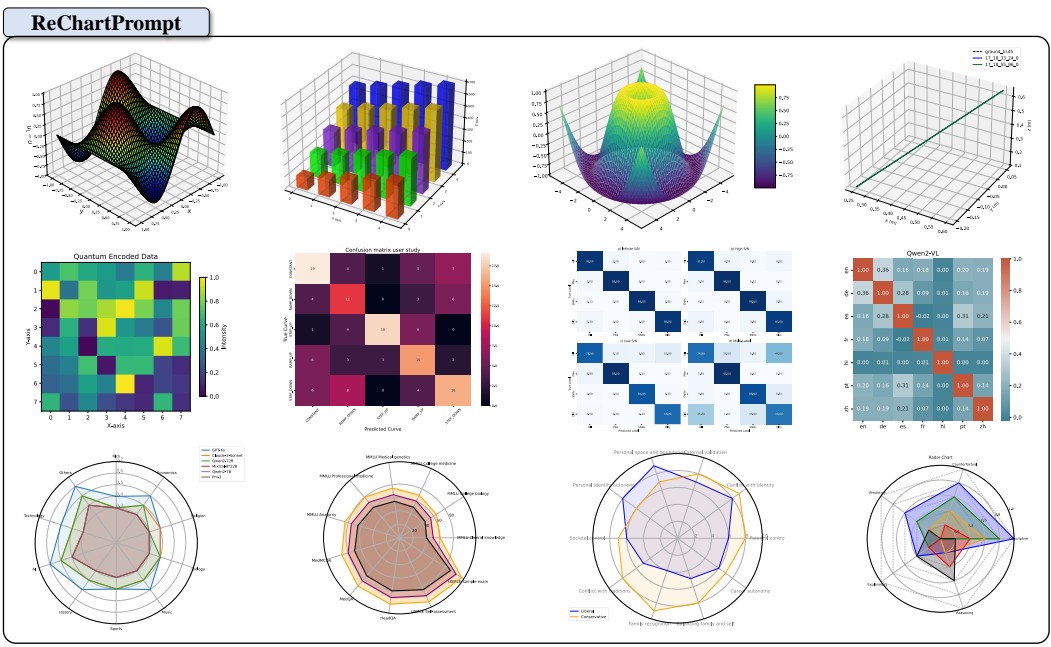

Figure 5: Dataset visualization. The charts in Chart2Code-160K exhibit homogenization, which affects diversity; the charts in ReChartPrompt demonstrate greater variety, especially in terms of textual content within the tables and layout attributes.

**$P_{type}$**

**You are given an image that represents one type of chart or plot. The possible plot types are:**

**[Bar, Line, ErrorBar, Heatmap, Box, Scatter, Hist, Radar, 3D, Pie, ErrorPoint, Violin]**

**Please carefully examine the given image and identify which one of the above plot types it belongs to.**

**- If the image clearly matches one of the plot types, respond with the exact name of that plot type (choose only one).**
**- If the image does not belong to any of these categories or is not a plot, respond with: None**

**Your answer should be exactly one word from the list above or None, nothing else.**

Figure 6: Prompt used for chart type classification ($P_{type}$). The Qwen2.5-VL-72B model is prompted with this template to assign each image to one of 12 predefined chart categories.

**$P_{rechart}$**

"You are an expert Python developer specializing in matplotlib. Based on the picture I provide, please write Python code using matplotlib to precisely reproduce the image.",
"As a skilled Python programmer with matplotlib expertise, please generate Python code that recreates the given image exactly.",
"You're an experienced matplotlib developer. Given the picture below, please write Python code that recreates it faithfully.",
"Please act as a Python matplotlib specialist and generate the Python code that reproduces the image shown below.",
"You are an expert in Python plotting using matplotlib. Create Python code to generate a plot identical to the provided picture.",
"Your task is to write matplotlib Python code that perfectly replicates the given image.",
"Imagine you are an expert Python coder who can write matplotlib code to duplicate images. Please generate code that reproduces the picture exactly.",
"You are requested to produce Python code using matplotlib that recreates the image below as closely as possible.",
"As a professional matplotlib developer, write Python code to visualize the given image precisely.",
"Please generate Python matplotlib code to recreate the picture shown.",
"You are a helpful assistant who can generate Python code using matplotlib. Please produce code to create a plot that closely resembles the given image, enclosed within ```python and ```.",
"You are a knowledgeable assistant specializing in matplotlib. Generate Python code that recreates the provided plot as closely as possible. The code should be wrapped in ```python and ```.",
"As a matplotlib expert assistant, please generate Python plotting code that replicates the given image. Output your code between ```python and ```.",
"You are a helpful bot that writes matplotlib Python code. Please provide the code to produce a plot that matches the image, wrapped in ```python and ```.",
"Your task is to produce matplotlib Python code that draws a plot visually similar to the given image. Enclose your code in ```python and ```.",
"You are a Python coding assistant with matplotlib skills. Please write code surrounded by ```python and ``` that recreates the given plot as closely as possible.",
"As an assistant proficient in matplotlib, generate Python code that reproduces the pictured plot. Your code should be enclosed in ```python and ```.",
"Generate Python matplotlib code that produces a plot similar to the provided image. Wrap the code inside ```python and ```.",
"You are an expert assistant that creates matplotlib Python code. Please write code enclosed in ```python and ``` that recreates the given picture as faithfully as possible.",
"Please generate Python code using matplotlib to produce a plot matching the given image, wrapped by ```python and ```. "

Figure 7: Prompt for chart-to-code generation ($P_{rechart}$). Twenty diverse prompts are designed to instruct the Qwen2.5-VL-72B model to generate Python matplotlib code from chart images, enhancing instruction diversity.

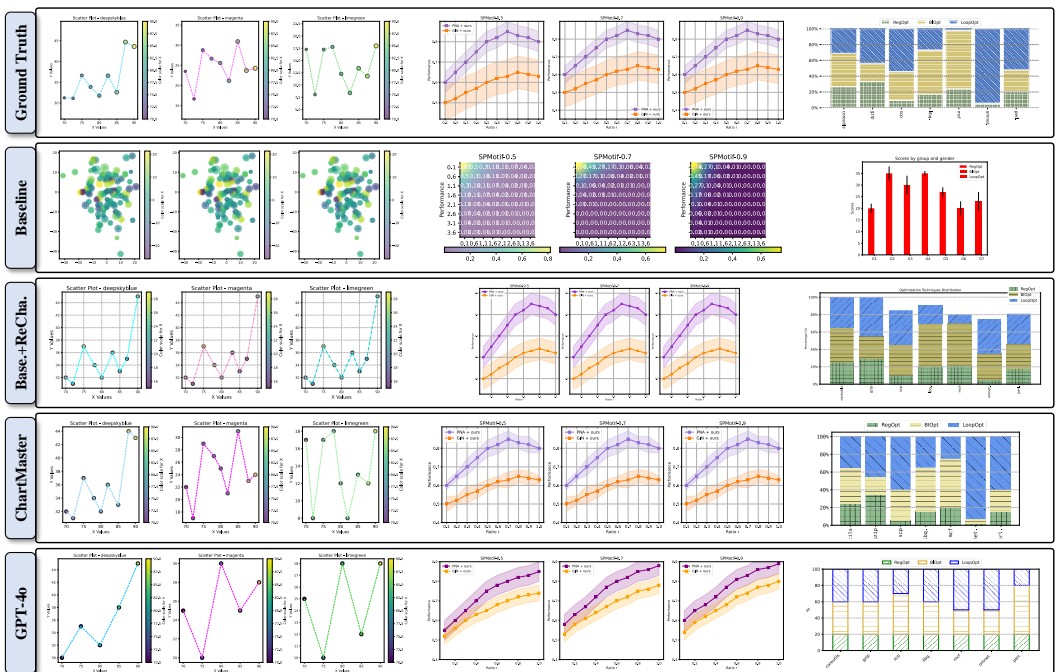

Figure 8: More test results of various models on the ChartMimic benchmark.

# A  DATA AUDITING

We provide a comprehensive summary of dataset statistics across each stage of data curation:

Table 6: Summary of Dataset Auditing Statistics by Data Curation Stage

| Stage | Statistic |
|---|---|
| Stage 1: Collecting Images from arXiv | Total images crawled: 477,788 |
| Stage 2: Filtering Non-Chart Images | Total chart images filtered: 288,992 |
| Stage 2: Chart-type misclassification rate | 44 / 1000 = 4.4% (manual check, 1000 samples from filtered charts) |
| Stage 3: Generating code with ReChartPrompt | Valid training samples (charts with executable code): 242,479 |
| Stage 4: Code Execution and Filtering | Code execution pass rate: 242,479 / 288,992 = 83.9% |
| Stage 4: Inter-annotator checks | Not required; successful code execution ensures reliable chart-code pairing |

**1. Chart-type misclassification (Stage 2).** To assess data quality, we randomly sample 1,000 chart images after the initial filtering stage and manually check for chart-type misclassification. Among these, 44 images (4.4%) contain both chart and non-chart elements but are classified as charts. In the subsequent Stage 4, only one of these 44 misclassified images successfully generates executable code; the remaining samples typically fail due to referencing non-existent files or incomplete code. Thus, the vast majority of noise is filtered out. Some examples can be found in Figure A of the supplementary materials.

**2. Inter-annotator checks (Stage 4).** Our pipeline relies on automatic code execution for validation, so manual inter-annotator agreement checks can be omitted.

**3. Style-consistent reproductions.** We clarify that our pipeline does not require replotted charts to be visually identical or style-consistent to the originals. Instead, the original charts are used to inspire diverse outputs, while code executability ensures each chart-code pair is valid. This design makes our dataset both diverse and reliable, while also simplifying the pipeline and reducing manual effort.

**4. Judging noise, coverage, and bias.** ReChartPrompt leverages real-world chart images from arXiv, resulting in diverse distributions and rich attribute coverage. Our dataset includes a wide variety of content and visual styles, which helps reduce bias and increase coverage compared to previous datasets. Code execution filtering further minimize noise. Figures 3 and 5 in the paper illustrate the attribute diversity and coverage of our dataset.

# B  IMPLEMENTATION AND EVALUATION DETAILS

| | ChartMiMic | Plot2Code | ChartX |
|---|---|---|---|
| **Prompt** | You are an expert Python developer who specializes in writing matplotlib code based on a given picture. I found a very nice picture in a STEM paper, but there is no corresponding source code available. I need your help to generate the Python code that can reproduce the picture based on the picture I provide. Note that it is necessary to use figsize=(X, Y) to set the image size to match the original size. Now, please give me the matplotlib code that reproduces the picture below. | You are a helpful assistant that can generate Python code using matplotlib.Generate the matplotlib code to create a plot that looks like the given image, as similar as possible.The generated code should be surrounded by \`\`\`python and \`\`\` | Redraw the chart image using Python code. |
| **Decoding Parameters** | context_length: 4096
max_tokens: 4096
temperature: 0.1
top_p: 1 | context_length: 4096
max_tokens: 4096
temperature: 0.1
top_p: 1 | context_length: 4096
max_tokens: 4096
temperature: 0.1
top_p: 0.9 |

Figure 9: Test prompts and decoding settings of benchmarks.

During the collection of arXiv papers, we explicitly exclude papers that are used as benchmarks to avoid potential data leakage. In the data generation stage, we apply a greedy sampling strategy to filter chart data, retaining only images of 12 predefined chart types and discarding all others. Then, we randomly select an instruction from $\mathbf{P}_{rechart}$ to prompt the Qwen2.5-VL-72B (Bai et al., 2025) model to generate code via nucleus sampling, with a temperature of 0.1 and a top-p of 0.9.

For training, we use the Qwen2.5-VL-7B model (Bai et al., 2025) in two stages. In Stage 1, we perform SFT on the entire ReChartPrompt-240K dataset with a learning rate of $2 \times 10^{-5}$, batch size 128, and a cosine annealing scheduler for one epoch; the resulting model is saved for Stage 2.

In Stage 2, ChartSimRL training is conducted on 10% of the ReChartPrompt-240K dataset, using a smaller learning rate of $5 \times 10^{-6}$ and generating $M = 4$ candidate codes per sample. Candidate sampling uses temperature 1.0, top-p 1.0, and top-k 80 to encourage diversity. The batch size remains 128 (32 samples $\times$ 4 candidates each).

For evelation, we assess the model's chart-to-code generation performance on multiple benchmarks. **ChartMimic Direct Mimic Task** (Yang et al., 2024a): This benchmark includes 600 chart images. GPT-4o scores (0–100) serve as high-level similarity metrics. Additionally, low-level F1 scores for text, layout, chart type, and color are computed from code execution for fine-grained analysis. **Plot2Code Direct Asking** (Wu et al., 2024): Metrics include code pass rate, text match rate, and a 10-point GPT-4V visual similarity score, jointly assessing code correctness and visual fidelity. **ChartX Chart Redrawing Task** (Xia et al., 2024): This benchmark uses GPT-4 (0–5 scale) to evaluate code-generated chart redrawings. The Test prompts and decoding settings are listed in Figure 9.

## C  STANDARD METRICS

We consider two RGB images: the original chart image $I_i \in \mathbb{R}^{H \times W \times 3}$ and the generated chart image $\hat{I}_i \in \mathbb{R}^{H \times W \times 3}$, where $H$ and $W$ denote the height and width of the images respectively (both images are resized to the same height and width before comparison), and 3 corresponds to the RGB color channels. Below, we describe how to quantify the visual similarity between $I_i$ and $\hat{I}_i$ using metrics such as Mean Squared Error (MSE), Structural Similarity (SSIM) (Wang et al., 2004), and Peak Signal-to-Noise Ratio (PSNR) (Hore & Ziou, 2010).

### C.1  MEAN SQUARED ERROR

The Mean Squared Error (MSE) is defined as:

$$\text{MSE}(I_i, \hat{I}_i) = \frac{1}{H \times W \times 3} \sum_{h=1}^{H} \sum_{w=1}^{W} \sum_{c=1}^{3} \left( I_i(h, w, c) - \hat{I}_i(h, w, c) \right)^2$$

This formula computes the average squared difference between the pixel values of the two images over all spatial locations and color channels. A smaller MSE indicates higher similarity between $I_i$ and $\hat{I}_i$.

To convert the MSE into a similarity score, we define the MSE-based similarity as:

$$\text{MSE\_Similarity} = \frac{1}{1 + \text{MSE}(I_i, \hat{I}_i)} \in (0, 1]$$

- When $\text{MSE} \to 0$, $\text{MSE\_Similarity} \to 1$, indicating the images are almost identical.
- When $\text{MSE} \to \infty$, $\text{MSE\_Similarity} \to 0$, indicating large differences between the images.

### C.2  STRUCTURAL SIMILARITY

The Structural Similarity (SSIM) is a perceptual metric that quantifies the similarity between two images by comparing local patterns of pixel intensities. It is computed on local sliding windows centered at each pixel location. For each window, local statistics including mean, variance, and covariance are calculated to evaluate the similarity. The final SSIM value for each channel is obtained by averaging these local SSIM values over all spatial positions, and the overall SSIM between two RGB images is computed by averaging over the three color channels.

Formally, for each color channel $c \in \{R, G, B\}$, the SSIM is defined as:

$$\text{SSIM}_c(I_i^c, \hat{I}_i^c) = \frac{(2\mu_{I_i^c}\mu_{\hat{I}_i^c} + C_1)(2\sigma_{I_i^c\hat{I}_i^c} + C_2)}{(\mu_{I_i^c}^2 + \mu_{\hat{I}_i^c}^2 + C_1)(\sigma_{I_i^c}^2 + \sigma_{\hat{I}_i^c}^2 + C_2)}$$

where

- $\mu_{I_i^c}$ and $\mu_{\hat{I}_i^c}$ are the local means computed within the sliding window.

- $\sigma_{I_i^c}^2$ and $\sigma_{\hat{I}_i^c}^2$ are the local variances.

- $\sigma_{I_i^c \hat{I}_i^c}$ is the local covariance.

- $C_1 = (K_1 L)^2$ and $C_2 = (K_2 L)^2$ are constants to stabilize the division, with default values $K_1 = 0.01$, $K_2 = 0.03$. $L$ is the dynamic range of the pixel values. For 8-bit grayscale images, $L = 255$. In our implementation, all images are converted to `np.float32` and normalized by dividing by 255, so the pixel values are in the range $[0, 1]$. Therefore, $L = 1.0$ is used for SSIM calculation.

The overall mean SSIM between the two RGB images is then calculated by averaging over all spatial positions $(x, y)$ in each channel and then over the three channels:

$$\text{SSIM}(I_i, \hat{I}_i) = \frac{1}{3} \sum_{c=1}^{3} \frac{1}{H \times W} \sum_{x=1}^{H} \sum_{y=1}^{W} \text{SSIM}_c(I_i^c(x, y), \hat{I}_i^c(x, y)) \in [0, 1]$$

- When $\text{SSIM} \to 1$, the images are structurally almost identical.
- When $\text{SSIM} \to 0$, there are significant structural differences between the images.

### C.3 PEAK SIGNAL-TO-NOISE RATIO

Peak Signal-to-Noise Ratio (PSNR) is a widely used metric to measure the quality of reconstructed images compared to the original images. It is defined as:

$$\text{PSNR}(I_i, \hat{I}_i) = 10 \log_{10} \left( \frac{L^2}{\text{MSE}(I_i, \hat{I}_i)} \right)$$

where $L$ is the dynamic range of the pixel values. For normalized images in $[0, 1]$, $L = 1.0$.

In practical scenarios, PSNR values typically range in tens of decibels and can vary widely, which may cause instability during optimization. To mitigate this effect, we normalize the PSNR values within each rollout batch by dividing them by the maximum PSNR value in that batch:

$$\text{PSNR}_{\text{norm}}(I_i, \hat{I}_i) = \frac{\text{PSNR}(I_i, \hat{I}_i)}{\max_{\hat{I}_j \in \text{rollout batch}} \text{PSNR}(I_j, \hat{I}_j)} \in (0, 1]$$

- When $\text{PSNR}_{\text{norm}} \to 1$, the reconstructed image $\hat{I}_i$ is very similar to the original image $I_i$.
- When $\text{PSNR}_{\text{norm}} \to 0$, there exist significant differences between the images.

## D THE USE OF LARGE LANGUAGE MODELS

In this study, the initial draft, core research ideas, motivation, data analysis, and scientific insights were all independently developed by the human authors. LLMs were used solely as auxiliary tools to polish the language of the initial draft, including removing redundant content and avoiding ambiguity, thereby enhancing the overall readability of the manuscript.

## E MORE EXPERIMENTS

### E.1 FINE-GRAINED RESULTS AND THEIR RELATIONSHIP WITH $R_i^{\text{attr}}$ AND $R_i^{\text{vis}}$

We provide a detailed quantitative breakdown to clarify how each reward component affects chart reconstruction quality. As shown in the Table 7, SFT on ReChartPrompt significantly boosts all metrics, laying a strong foundation. Adding either the attribute reward or visual similarity reward further improves low-level metrics, but their effects differ.

Specifically, $R_i^{\text{attr}}$ mainly enhances text accuracy and layout fidelity, but has limited impact on color consistency. This is because $R_i^{\text{attr}}$ relies on discrete matching, where both subtle and large color

Table 7: Fine-grained quantitative analysis on ChartMimic benchmark.

| ReChartPrompt | ChartSimRL $R_i^{\text{attr}}$ | $R_i^{\text{vis}}$ | Exec. Rate | Low-Level | | | | | High-Level GPT-4o |
|---|---|---|---|---|---|---|---|---|---|
| | | | | Text | Layout | Type | Color | Avg. | |
| | | | 65.5 | 35.2 | 58.1 | 37.8 | 28.3 | 39.9 | 40.7 |
| ✓ | | | 91.1 | 75.6 | 87.8 | 67.0 | 64.3 | 73.7 | 80.9 |
| ✓ | ✓ | | 92.1 | 80.1 | 90.2 | 69.5 | 65.1 | 76.2 | 83.9 |
| ✓ | | ✓ | 92.1 | 79.8 | 90.6 | 71.8 | 68.7 | 77.7 | 84.3 |
| ✓ | ✓ | ✓ | 93.8 | 79.8 | 91.3 | 72.2 | 69.7 | 78.2 | 85.1 |

differences are treated as mismatches, even though larger discrepancies should be penalized more heavily. In contrast, the visual similarity reward ($R_i^{\text{vis}}$), which evaluates global image features in a continuous manner, better captures approximate color and gradient variations, resulting in stronger gains in color consistency.

Therefore, the optimal approach is to combine both reward mechanisms, leveraging their complementary strengths to achieve robust and fine-grained chart-to-code reconstruction.

## E.2 EXTENSION TO CHART UNDERSTANDING TASKS

Table 8: Ablation study on the impact of ReChartPrompt data and ChartSimRL for chart understanding tasks.

| Tiny Chart | ReChart Prompt | SFT | RL | ChartQA | ChartQAPro | | | | | |
|---|---|---|---|---|---|---|---|---|---|---|
| | | | | | Factoid | Conversational | Hypothetical | Fact Checking | Multi Choice | Overall |
| ✓ | | ✓ | | 87.8 | 26.7 | 39.7 | 41.7 | 38.5 | 35.5 | 34.2 |
| ✓ | ✓ | ✓ | | 89.2 | 27.5 | 42.1 | 36.0 | 45.0 | 39.2 | 37.9 |
| ✓ | ✓ | ✓ | ✓ | 89.8 | 29.3 | 43.4 | 38.9 | 47.1 | 36.9 | 39.1 |

To verify the effectiveness of our method on chart understanding tasks, we conduct further experiments on ChartQA (Masry et al., 2022) and ChartQAPro (Masry et al., 2025a) benchmarks.

Following ChartCoder, we incorporate the TinyChart dataset (Zhang et al., 2024b) throughout the training process. Specifically, we first use 240K TinyChart instances for SFT on Qwen2.5-VL-7B as the baseline. Then, we jointly train the model with our own dataset during SFT. During GRPO, we use 24K QA samples from ChartQA (Masry et al., 2022), PlotQA (Methani et al., 2020), and DVQA (Kafle et al., 2018) subsets, applying an accuracy-based reward for QA and attribute/visual rewards for chart-to-code. Losses for QA and chart-to-code tasks are computed separately and averaged; other hyperparameters remain unchanged.

As shown in Table 8, incorporating ReChartPrompt data during SFT notably improves QA accuracy, especially for Fact Checking, with further gains from RL. This demonstrates that chart-to-code learning enhances the model's fine-grained understanding of chart semantics and transfers effectively to reasoning tasks, resulting in better QA performance.

## E.3 IMPACT OF TEACHER MODEL

Table 9: Impact of teacher model quality on ChartMaster performance.

| Method | SFT | GRPO | ChartMimic | | | Plot2Code | | | ChartX |
|---|---|---|---|---|---|---|---|---|---|
| | | | Exec.Rate | Low-Level | High-Level | Pass Rate | Text-Match | Rating | GPT-score |
| Qwen2.5-VL-7B | | | 65.5 | 39.9 | 40.7 | 67.4 | 43.8 | 4.60 | 2.18 |
| Qwen2.5-VL-72B | | | 88.5 | 72.7 | 79.1 | 84.8 | 68.4 | 6.83 | 2.52 |
| Qwen3-VL-235B-A22B-Instruct | | | 94.0 | 79.1 | 82.3 | 90.1 | 56.3 | 6.49 | 2.94 |
| Use Qwen2.5-VL-72B | ✓ | | 91.1 | 73.7 | 80.9 | 80.3 | 59.3 | 5.34 | 2.36 |
| as Teacher Model | ✓ | ✓ | 93.8 | 78.2 | 85.1 | 88.2 | 62.6 | 5.65 | 2.46 |
| UseQwen3-VL-235B | ✓ | | 91.1 | 75.3 | 81.5 | 82.5 | 64.3 | 5.57 | 2.48 |
| as Teacher Model | ✓ | ✓ | 95.1 | 79.4 | 86.2 | 88.6 | 65.7 | 5.93 | 2.53 |

To investigate the impact of the teacher model on ChartMaster performance. We select Qwen3-VL-235B-A22B-Instruct as the stronger teacher to generate a new 240K chart-to-code dataset. As shown in Table 9, ChartMaster's performance improves significantly when a stronger teacher model is used for data distillation, demonstrating that teacher quality substantially impacts student performance. Importantly, our method enables the student model to closely match and even outperform the teacher

on certain metrics, evidencing the effectiveness of our approach in leveraging high-quality teacher knowledge.

## E.4 ROBUSTNESS OF VISUAL REWARD TO CHART SEMANTICS

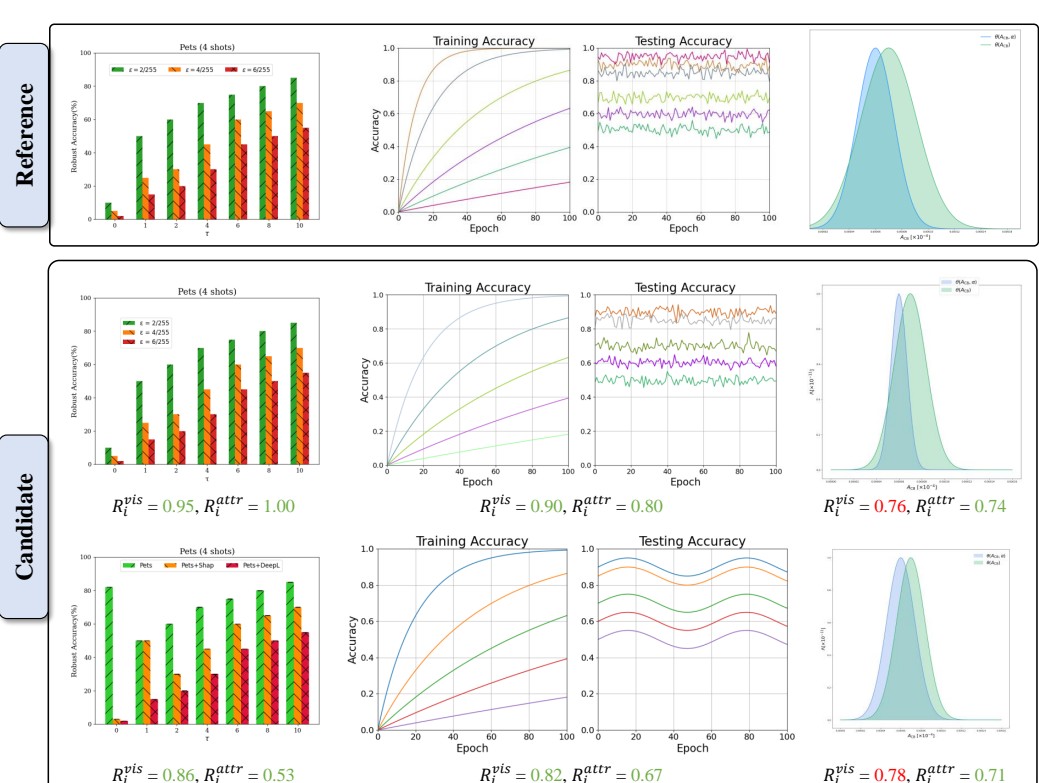

Figure 10: Qualitative analysis of candidate charts generated during GRPO. Visual reward and attribute (semantic) reward are generally positively correlated. Outlier cases with high visual reward but low semantic alignment receive low final reward, indicating that our design avoids overfitting to style surrogates.

To further examine whether our visual reward overfits to style surrogates, we conduct a qualitative analysis of candidate charts generated during the GRPO process. We observe in Figure 10 that nearly all candidates with high visual scores also achieve high attribute (semantic) scores, indicating strong semantic alignment. Occasionally, some candidates exhibit high visual scores but low attribute scores; in these cases, the final reward remains low due to the penalization from the attribute score. These results suggest that the visual reward does not cause overfitting to superficial styles, and the attribute score effectively mitigates the impact of outliers.

## E.5 ERROR ANALYSIS

We conduct error analysis on ChartMaster-7B using the ChartMimic test set and present typical failure cases in Appendix Figure 11. The results reveal that the primary source of error is the inaccurate extraction of precise numerical values from complex charts. Despite implementing a relaxed matching strategy for numerical values, this issue remains unresolved. Further exploration of reward design and model architecture will be pursued in future work.

## E.6 THE IMPACT OF CHART-AWARE VISUAL ENCODER

In Table 5, we have compared the performance of different CNN-based visual encoders and concluded that deep visual features are effective. To further investigate, we use the chart-aware visual encoder from ChartCoder-7B to extract features. As shown in Table 10, the chart-aware encoder

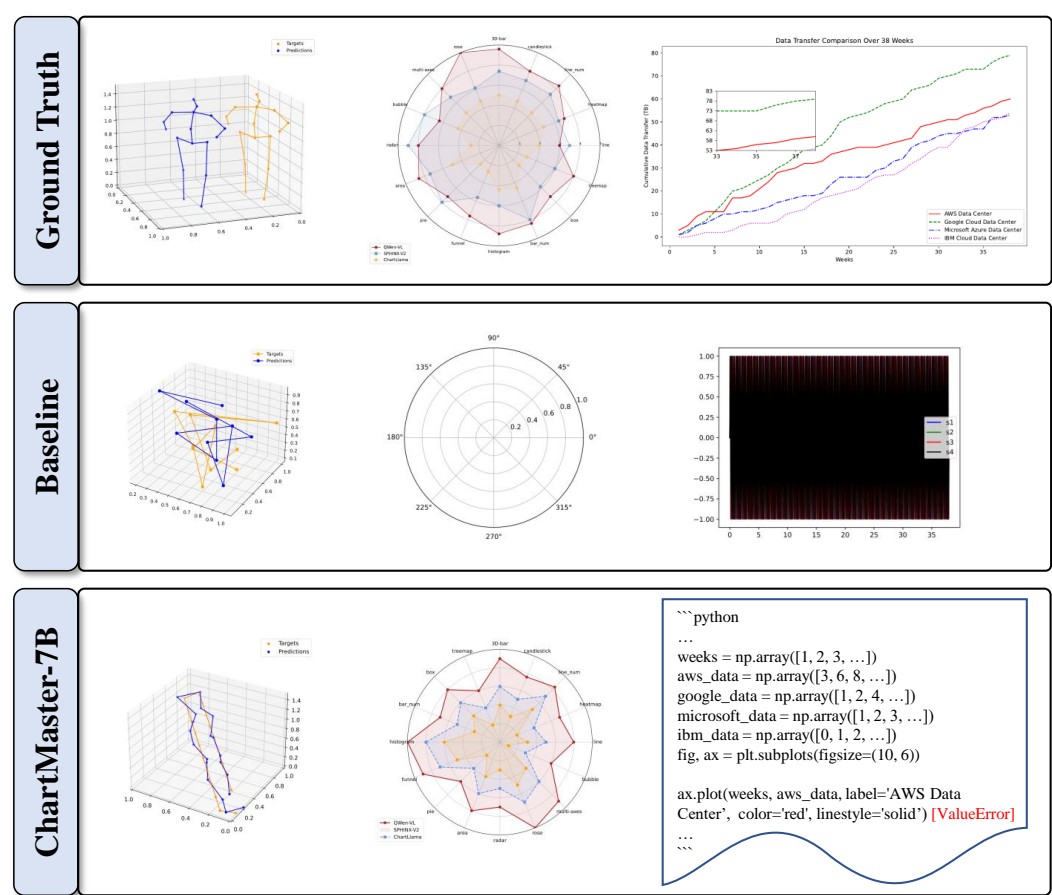

Figure 11: Some bad cases from ChartMaster-7B on the ChartMimic test set. The main challenge lies in accurately extracting precise numerical values.

Table 10: Comparison of ChartMaster-7B performance using different visual encoders on the Chart-Mimic benchmark.

| ChartMaster-7B | Exec. Rate | Low-Level | | | | | High-Level |
|---|---|---|---|---|---|---|---|
| | | Text | Layout | Type | Color | Avg. | GPT-4o |
| w/ ChartCoder-ViT | 94.8 | 83.1 | 93.1 | 71.1 | 65.7 | 78.2 | 84.6 |
| w/ ResNet-18 | 93.8 | 79.8 | 91.3 | 72.2 | 69.7 | 78.2 | 85.1 |

slightly improves execution rate and achieves better text and layout metrics, reflecting enhanced chart-specific feature extraction. However, it lags behind ResNet-18 in Type and Color metrics, suggesting that traditional CNNs may better capture texture and color information. Overall, both encoders show comparable average performance. We will continue exploring more specialized visual encoders in future work.

## E.7 PROMPT ANALYSIS

Table 11: Comparison of model performance trained on single-prompt versus diverse-prompt datasets. Models trained with diverse prompts generally achieve higher scores, illustrating the benefit of prompt diversity.

| Dataset | ChartMimic | | | Plot2Code | | | ChartX |
|---|---|---|---|---|---|---|---|
| | Exec.Rate | Low-Level | High-Level | Pass Rate | Text-Match | Rating | GPT-score |
| ReChartPrompt-240K-Single-Prompt | 88.5 | 73.0 | 78.9 | 83.3 | 58.9 | 5.14 | 2.30 |
| ReChartPrompt-240K-Diverse-Prompt | 91.1 | 73.7 | 80.9 | 80.3 | 59.3 | 5.34 | 2.36 |

We investigate the impact of prompt diversity on code generation quality. Specifically, we randomly select 1,000 original charts and generate replotted results using each of the 20 prompts in $\mathbf{P}_{\text{rechart}}$. The code execution rates are comparable across prompts: 82.7%, 84.2%, 81.9%, 81.5%, 83.8%, 82.9%, 82.1%, 82.3%, 83.2%, 84.0%, 81.8%, 82.4%, 84.1%, 82.7%, 83.6%, 81.2%, 84.4%, 81.5%, 84.7%, and 83.3%. This indicates that Qwen2.5-VL-72B demonstrates strong instruction-following ability, and different phrasings of similar prompts yield no significant differences in code pass rates.

To further assess the effect of prompt diversity, we identify the prompt with the highest code pass rate and use it to regenerate 240K training samples (ReChartPrompt-240K-Single-Prompt). We then compare these results to those obtained from our diverse prompt dataset (ReChartPrompt-240K-Diverse-Prompt). As shown in Table 11, models trained with diverse prompts consistently outperform those trained with a single prompt across multiple benchmarks, demonstrating the clear advantage of prompt diversity in improving model performance.

