# SUPPLEMENTARY MATERIALS

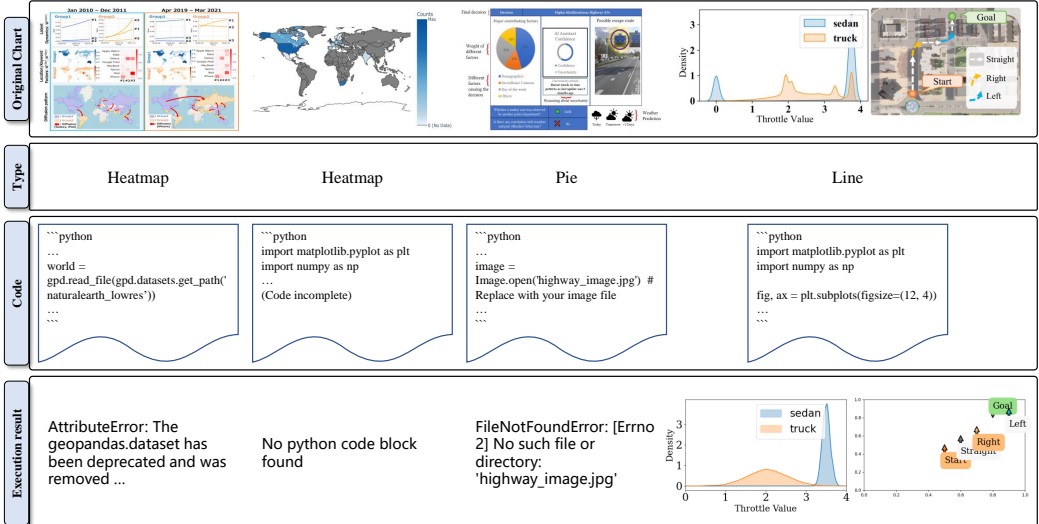

Figure A: Examples of misclassified chart images. Each column shows: the original image (Higashiguchi et al., 2025; Olson et al., 2025; Mehrotra et al., 2025; Shin et al., 2024), assigned chart type, extracted code, and code execution result. Most misclassified cases either fail to execute due to missing files or code errors, ensuring they are filtered out at the code validation stage.

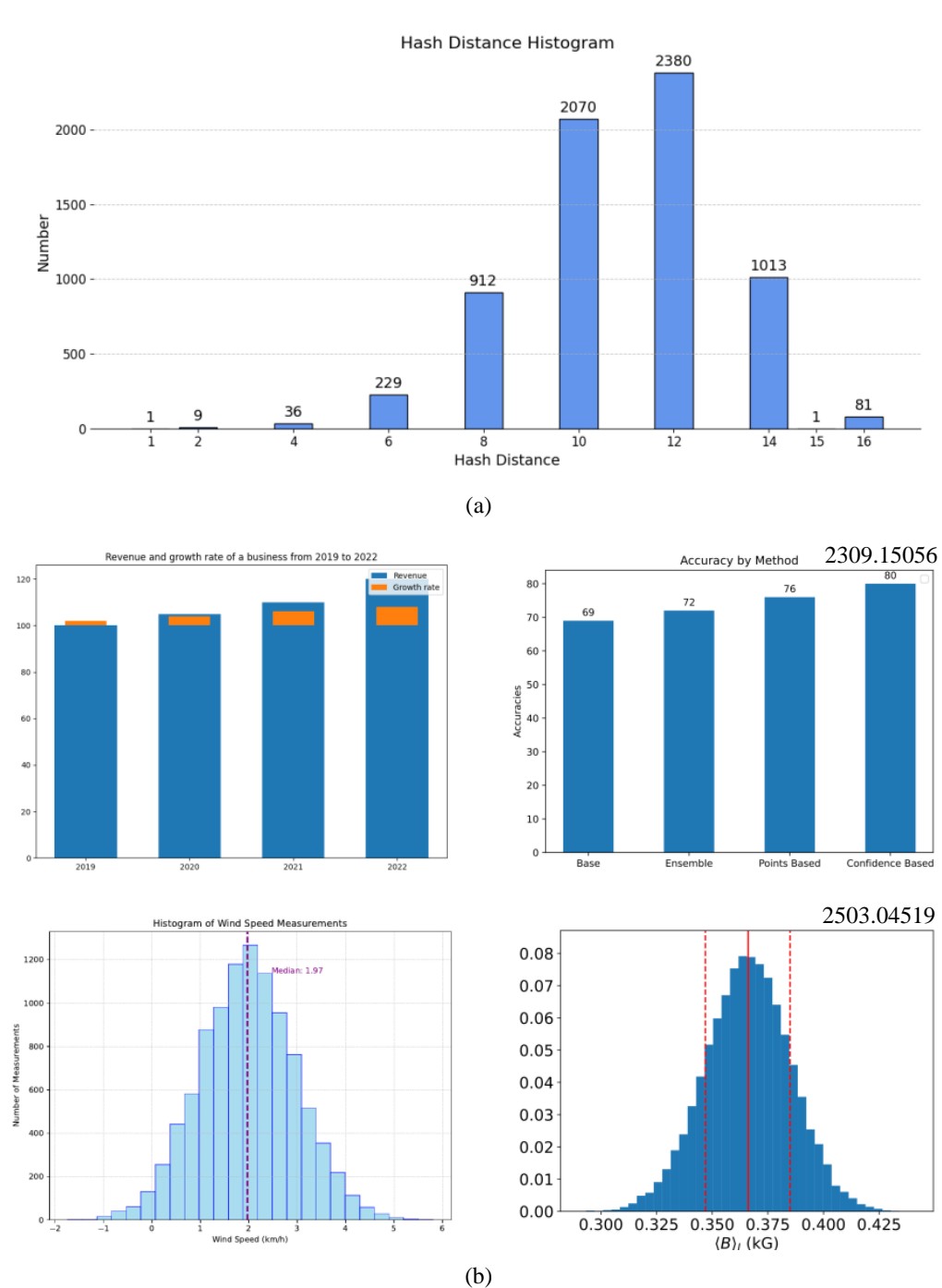

(a)

(b)

Figure B: (a) Histogram of minimum hash distances between each test image and its most similar chart (Silver et al., 2022; Hahlin et al., 2025) in the 288,992 charts set, indicating no near-duplicate images. (b) Examples of test images and their closest charts with hash distance 1–2, showing that even the most similar samples are visually distinct.

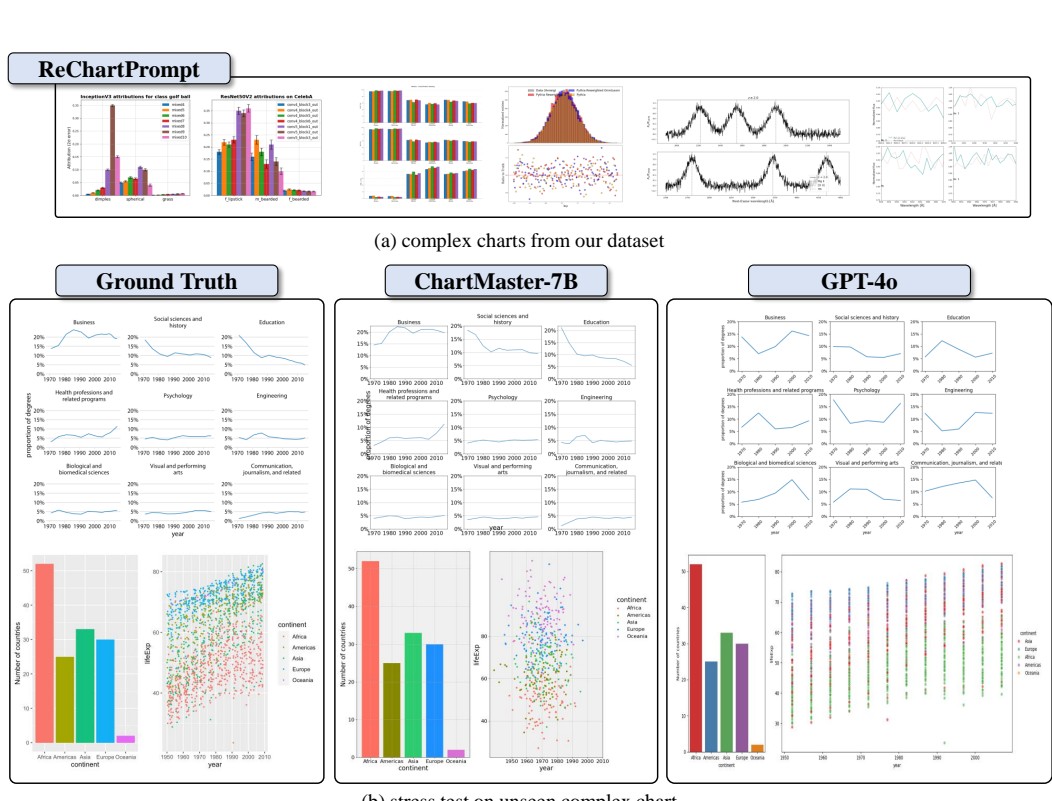

(a) complex charts from our dataset

(b) stress test on unseen complex chart

Figure C: (a) Examples of complex charts in the ReChartPrompt-240K dataset, illustrating the diversity and difficulty beyond simple chart types. (b) Stress test results on manually collected complex charts. ChartMaster-7B demonstrates strong generalization and visual consistency, successfully reconstructing multi-panel and faceted charts, and outperforming GPT-4o in structure fidelity.