# OpenReview forum: "ChartMaster: Advancing Chart-to-Code Generation with Real-World Charts and Chart Similarity Reinforcement Learning"
_ICLR.cc/2026/Conference — Submitted to ICLR 2026_

### Official Review · Reviewer_ER2g · 2025-10-31

**Soundness:** 3
**Presentation:** 3
**Contribution:** 3
**Rating:** 6
**Confidence:** 4

**Summary:**

Introduced “ReChartPrompt-240K”, a new real-world chart-to-code dataset with higher variability.
Introduced ChartSimRL, a GRPO-based RL algorithm to ensure the code-generated images align with the visual and the semantic attributes of the input image.

**Strengths:**

The first model to focus on addressing the Chart-To-Image challenges.
Comparison to many of the known base language models.
Comprehensive and fine-grained ablation studies justifying almost every decision.
Inclusion of RL training tailored for visual structure and semantic similarities.
Introduces a new dataset with higher real-world varieties.
Well presented.
SOTA results compared to other similar and larger models.

**Weaknesses:**

Qwen2.5-VL-72B was used to filter the non-chart images, but this approach was not validated. What if some images are categorized as charts when they are not? Yes, the dataset achieved better results at the end, which indicates that it worked, but this may need further validations; maybe another tailored way of classification would enhance the introduced dataset.

No explanation of how the introduced semantic attribute extraction tool works, it is mentioned that it extracts the colors, numbers, texts, and the layout, but it’s not mentioned how exactly this is done from the chart image, and if the extraction is validated somehow.

**Questions:**

Have you done any prompt-based analysis? You used 20 prompts for the chart generation. Do some prompts persistently produce better results?

**Details Of Ethics Concerns:**

They used the arXiv API to download the papers and used the images from 30,071 different papers, but they didn’t mention whether they had validated under which license those papers were published.

As far as I researched, the arXiv content, like images, is not necessarily published under a license that allows reusing the images in this manner. However, most of the papers might be published under a relaxed license that allows doing so, but this might need to be reviewed to make sure the used papers are published under the correct license, especially since they are planning to publish their constructed dataset (From the downloaded papers' images) as an open source dataset..

---

> ### Author Response · Authors · 2025-11-23
>
> **Q1**: Chart filtering validation.
>
> **A1**: Thank you for your insightful comment. To address this concern, we conduct an additional validation and include the results in Figure A of the supplementary materials.
>
> Specifically, we randomly sample 1,000 images labeled as charts after the initial filtering stage and manually check for misclassification. Among these, 44 images contain both chart and non-chart elements but are classified as charts. In the code execution and filtering stage, only one of these 44 misclassified images successfully generates executable code; the remaining samples typically fail due to referencing non-existent files or incomplete code and are excluded from the training set. This demonstrates that the vast majority of noise is filtered out by our pipeline. Therefore, as shown in Table 2, our dataset significantly improves model performance on the chart-to-code generation benchmarks.
>
> **Q2**: Semantic attribute extraction tool.
>
> **A2**: Thank you very much for your valuable feedback. Our semantic attribute extraction tool is built upon the attribute extraction code from the ChartMimic benchmark. The tool works by executing the model-generated code to obtain a matplotlib object, from which we extract member attributes to obtain specific chart properties, as illustrated in Figure 2 (c). During development, we also performed manual validation to ensure that the extracted attributes accurately and comprehensively reflect the chart content. The reliability of our tool is further supported by experimental results in Tables 2, 3, and 4, which demonstrate its effectiveness and its contribution to improved model performance. To ensure full reproducibility, we plan to release all related code, data, and models, including the implementation of this extraction tool.
>
> **Q3**: Prompt analysis.
>
> |               Dataset                             |    \|ChartMimic      ||        |   \|Plot2Code         |  |    |  \|ChartX  |
> |-------------------------------------------|------------|-----------|------------|-----------|------------|--------|-----------|
> |                                           |     \|Exec.Rate | Low-Level | High-Level | \|Pass Rate | Text-Match | Rating | \|GPT-score |
> | ReChartPrompt-240K-Single-Prompt    | \|88.5                 | 73.0                  | 78.9                  | \|83.3                 |58.9                  |  5.14              | \|2.30                |
> | ReChartPrompt-240K-Diverse-Prompt    | \|91.1                 |  73.7                  |  80.9                  | \|80.3                | 59.3                   |  5.34            | \|2.36               |
>
> **A3**: Thank you for this insightful question. Our main motivation for using multiple prompts is to introduce instruction diversity. The original chart acts as a visual cue to encourage varied outputs. Notably, we do not require replotted charts to closely match the originals. However, we do expect the replotted chart and code to be aligned, and the execution filtering process ensures this.
>
> Therefore, we can evaluate the effectiveness of different prompts by analyzing the code pass rates for each prompt. To this end, we randomly select 1,000 original charts and generate replotted results using each of the 20 prompts in $\mathbf{P}_{\text{rechart}}$. The code execution rates are comparable across prompts: 82.7\%, 84.2\%, 81.9\%, 81.5\%, 83.8\%, 82.9\%, 82.1\%, 82.3\%, 83.2\%, 84.0\%, 81.8\%, 82.4\%, 84.1\%, 82.7\%, 83.6\%, 81.2\%, 84.4\%, 81.5\%, 84.7\%, and 83.3\%. This indicates that Qwen2.5-VL-72B demonstrates strong instruction-following ability, and different phrasings of similar prompts yield no significant differences in code pass rates.
>
> We also identify the prompt with the highest code pass rate and use it to regenerate 240K training samples (ReChartPrompt-240K-Single-Prompt), comparing them with our diverse prompt dataset (ReChartPrompt-240K-Diverse-Prompt). As shown in the table, models trained with diverse prompts consistently outperform those trained with a single prompt across multiple benchmarks, clearly demonstrating the advantage of prompt diversity.
>
> These results are included in Appendix E.7. Thank you!
>
> **Q4**: License.
>
> **A4**: Thank you for raising this important question. When collecting papers from arXiv, we carefully considered licensing issues. We only included papers published under permissive Creative Commons licenses (CC BY, CC BY-SA, CC BY-NC-SA, and CC0). These licenses allow content to be reused under certain conditions, such as non-commercial use.
>
> Furthermore, we do not release any original source charts. We only plan to release the code of ReChartPrompt-240K, as well as the execution scripts used to replot charts from this code. This ensures that our open-source dataset does not redistribute any original copyrighted material and fully complies with licensing requirements.

---

### Official Review · Reviewer_5icT · 2025-10-31

**Soundness:** 3
**Presentation:** 2
**Contribution:** 2
**Rating:** 4
**Confidence:** 4

**Summary:**

Authors propose ChartMaster, a chart to code system with two main parts: ReChartPrompt data pipeline that crawls charts (from arxiv) and uses an VLM to generate plotting code and ChartSimRL, a GRPO style RL stage that optimizes code with a combined reward: attribute similarity and visual similarity, plus execution signals. Their model is trained on Qwen2.5-VL 7B and reported to outperform open-source baselines and approach large closed models on ChartMimic, Plot2Code and ChartX.

**Strengths:**

- Clear algorithmic core is heart of the paper, where reward is explicitly defined and implementable (Jaccard on attributes + ResNet-18 cosine on multi-level features under a GRPO objective). This is a reasonable and task-aligned signal for chart reproduction.
- Competitive 7B results are posted. authors claims best open-source 7B performance across several chart-to-code benchmarks.. the table lists both closed and open models for context.
- Leakage awareness is studied to an exten. Authors state they excluded papers used as benchmarks during arXiv crawling to reduce leakage risk..good / recommended practice for this area

**Weaknesses:**

- my one concern is that their data construction lacks quantitative auditing. While the paper says it crawls real-world tables/charts and filters non-executable code, it doesn’t quantify data quality: e.g., % of images incorrectly chart-typed, code-render success rate after filtering, or inter-annotator checks. Without a data card–style audit, it’s hard to judge noise, coverage, and bias in ReChartPrompt.
- [minor] does reward overfit to style surrogates? their visual reward uses ResNet-18 features and cosine similarity averaged across blocks. this is simple and efficient, but may overweight low-level textures/layout while underweighting higher-order chart semantics (e.g., axis scaling quirks, legend mapping, tick formatting). some abalation evidence (even if qualitative) of failure cases or sensitivity analyses can make things stronger.
- a reproducibility appendix with exact prompts and decoding configs for every baseline you reran would help..since benchmarks ChartMimic, Plot2Code, ChartX are "filled by our own experiments"
-  intro/claims suggest "near GPT-4o performance with only 7B."  comparison table indeed shows strong 7B results. Please consider either (a) add paired, identical decoding for GPT-4o across all tasks/metrics you report, or perhaps (b) soften wording to "competitive among open-source 7B." to be fair.
- Excluding benchmark papers is good, but not sufficient: figures often recirculate across venues, tech blogs, and textbooks. if possible, kindly show a near-duplicate search (image hashing / CLIP retrieval) between train vs. eval images (and code) with collision rates to substantiate the no-leakage claim.
- Also, data collection retains only 12 predefined chart types. That keeps scope crisp, but may bias models to common/simple forms and limit transfer to bespoke charts (multi-panel, maps, complex faceting). A stress test on “unseen” or composite chart types would strengthen the story.
- [Minor] Missing error analysis and qualitative diagnostics.

missing references of some relevant papers on visual reasoning and visual RL:
[1] Masry et al. BigCharts-R1: Enhanced Chart Reasoning with Visual Reinforcement Finetuning, https://arxiv.org/abs/2508.09804
[2] Rodriguez et al, BigDocs: An Open Dataset for Training Multimodal Models on Document and Code Tasks. https://arxiv.org/abs/2412.04626
[3] Xia et al, StructChart: Perception, Structuring, Reasoning for Visual Chart Understanding.
[4] Awal et al. WebMMU: A Benchmark for Multimodal Multilingual Website Understanding and Code Generation https://arxiv.org/abs/2508.16763.

**Questions:**

- What is the code execution pass-rate after filtering? What % of crawled images end up with valid, style-consistent reproductions? Any manual spot-checks or annotator agreement?
- wondering how sensitive are results to the ResNet-18 backbone? What happens if you switch to a chart-aware visual encoder or add OCR-aware features to the reward?
- What is a crucial difference and possible advantage between this work and BigCharts-R1? To me it's nearly the same.


idea (attribute and visual rewards under GRPO)  and the 7B results look strong, but the paper may needs deeper data QA, leakage audits, and ablations to match top-tier standards. Addressing the points above would significantly strengthen the case. Happy to increase the scores during the response period.

---

> ### Author Response · Authors · 2025-11-23
> **Part 1/3**
>
> **Q1** :  Data auditing.
>
> **A1** :  Thank you for your thoughtful feedback regarding quantitative auditing.
>
> We provide a comprehensive summary of dataset statistics across each stage of data curation:
>
> | **Stage**                          | **Statistic**                                                                                                   |
> |-----------------------------------|----------------------------------------------------------------------------------------------------------------|
> | Stage 1: Collecting Images from arXiv        | Total images crawled: 477,788                                                                                  |
> | Stage 2: Filtering Non-Chart Images           | Total chart images filtered: 288,992                                                                            |
> | Stage 2: Chart-type misclassification rate    | 44 / 1000 = 4.4% (manual check, 1000 samples from filtered charts)                                             |
> | Stage 3: Generating code with ReChartPrompt   | Valid training samples (charts with executable code): 242,479                                                  |
> | Stage 4: Code Execution and Filtering          | Code execution pass rate: 242,479 / 288,992 = 83.9%                                                            |
> | Stage 4: Inter-annotator checks                 | Not required; successful code execution ensures reliable chart-code pairing                                     |
>
> ---
>
> **1. Chart-type misclassification (Stage 2).**
> To assess data quality, we randomly sample 1,000 chart images after the initial filtering stage and manually check for chart-type misclassification. Among these, 44 images (4.4%) contain both chart and non-chart elements but are classified as charts. In the subsequent Stage 4, only one of these 44 misclassified images successfully generates executable code; the remaining samples typically fail due to referencing non-existent files or incomplete code. Thus, the vast majority of noise is filtered out. Some examples can be found in Figure A of the supplementary materials.
>
> **2. Inter-annotator checks (Stage 4).**
> Our pipeline relies on automatic code execution for validation, so manual inter-annotator agreement checks can be omitted.
>
> **3. Style-consistent reproductions.**
> We clarify that our pipeline does not require replotted charts to be visually identical or style-consistent to the originals. Instead, the original charts are used to inspire diverse outputs, while code executability ensures each chart-code pair is valid. This design makes our dataset both diverse and reliable, while also simplifying the pipeline and reducing manual effort.
>
> **4. Judging noise, coverage, and bias.**
> ReChartPrompt leverages real-world chart images from arXiv, resulting in diverse distributions and rich attribute coverage. Our dataset includes a wide variety of content and visual styles, which helps reduce bias and increase coverage compared to previous datasets. Code execution filtering further minimizes noise. Figures 3 and 5 in the paper illustrate the attribute diversity and coverage of our dataset.
>
> We sincerely appreciate your feedback and hope these detailed auditing statistics address your concerns. The full data auditing results are added in Appendix A.
>
> **Q2** :  Robustness of visual reward to chart semantics.
>
> **A2** : Thank you for your insightful question. To assess whether our visual reward overfits to style surrogates, we analyze candidate charts from the GRPO process and compare them with their references. As shown in the newly added Appendix Figure 10, nearly all candidates with high visual scores also have high attribute (semantic) scores, meaning charts with higher visual scores closely match the original semantics. Rare samples have high visual scores but incorrect semantics; in these cases, the attribute score is low, which penalizes such samples and results in a low final reward. This shows that the visual reward itself does not lead to overfitting to style surrogates, while the attribute score further reduces the impact of outliers. It is worth noting that our visual reward improves Low-Level scores by 4 points and High-Level scores by 3.4 points on ChartMimic (Table 3), and Figure 4 shows that ChartMaster more accurately captures fine-grained visual details and key chart attributes, resulting in better ground truth reproduction than the baseline.

---

> ### Author Response · Authors · 2025-11-23
> **Part 2/3**
>
> **Q3** :  Reproducibility and fair benchmarking.
>
> |               Model                             |    \|ChartMimic      ||        |   \|Plot2Code         |  |    |  \|ChartX  |
> |-------------------------------------------|------------|-----------|------------|-----------|------------|--------|-----------|
> |                                           |     \|Exec.Rate | Low-Level | High-Level | \|Pass Rate | Text-Match | Rating | \|GPT-score |
> | GPT-4o report     | \|93.2                 | 79.0                  | 83.5                  | \|88.6                 | 56.3                  | 5.71              | \|-                 |
> | GPT-4o*           | \|93.5                 | 78.4                  | 83.6                  | \|90.1                 | 54.1                  | 5.76              | \|2.36              |
> | ChartMaster-7B    | \|93.8                 | 78.2                  | 85.1                  | \|88.2                 | 62.6                  | 5.65              | \|2.46              |
>
> **A3** : Thank you very much for your valuable comments.
> Our experiments use the official codebases with default prompts and decoding strategies, as detailed in the newly added Appendix Figure 9. Most results in Table 1 are cited from the original benchmark papers; for missing results, we conduct evaluations using the official settings shown in Figure 9, and mark these supplemented results with an asterisk ($^{*}$) in the revised paper. To ensure fair comparison, we rerun GPT-4o under the same settings. As shown in the table, the results are consistent with previously reported performance, and ChartMaster achieves performance comparable to GPT-4o. For full reproducibility, we will release all datasets, code, and models.
>
> **Q4** :  Evidence for no overlap.
>
> **A4** : To address potential data leakage, we apply image hashing to all 288,992 charts and 6,732 test images to identify near-duplicates, as image hashing is effective for large-scale datasets. For each test image, we find the chart with the smallest hash distance and manually check for exact matches; no identical images are found. Supplementary Figure B shows the distribution of minimum hash distances for all test images. We also manually inspect samples with hash distances of 1–2, further confirming the absence of duplicates. This analysis provides strong evidence that our dataset does not suffer from data leakage.
>
> **Q5** :  Generalization to composite charts.
>
> **A5** : Thank you for your suggestion. We define 12 chart types mainly to filter out non-chart images during data collection; this does not limit the model to simple forms. As shown in Supplementary Figure C (a), the ReChartPrompt-240K dataset contains many complex charts.
>
> To further test generalization, we manually collect several complex charts from Google for stress test. Supplementary Figure C (b) also shows that ChartMaster-7B successfully captures the main structure of multi-panel and faceted charts, with better visual consistency than GPT-4o. These results confirm our model’s strong generalization ability.
>
> **Q6** :  Error analysis.
>
> **A6** : Thank you. We present bad cases from ChartMaster-7B on the ChartMimic test set in Appendix Figure 11. The main challenge lies in accurately extracting precise numerical values from complex charts. Although we implement a relaxed matching strategy for numerical values (see Line 233), this issue persists. As a potential solution, we plan to introduce the mean squared error (MSE) of numerical attributes as an additional reward signal during training to further improve numerical accuracy. We will continue to explore and evaluate more effective solutions to address this challenge in future work.
>
> These results are added in Appendix E.5. Thank you!

---

> ### Author Response · Authors · 2025-11-23
> **Part 3/3**
>
> **Q7** :  Comparison with BigCharts-R1 and adding references.
>
> **A7** : Thank you for highlighting these important related works. We have added them in our revised paper. Below, we analyze the key differences between ChartMaster and BigCharts-R1 in terms of motivation, data, and reward design.
>
> **Motivation:**
> BigCharts-R1 aims to improve model QA capabilities by addressing the challenge of limited QA accuracy in diverse real-world QA datasets. In contrast, ChartMaster focuses on enhancing chart-to-code generation, tackling the scarcity and lack of diversity in existing chart-to-code datasets.
>
> **Data Pipeline:**
> BigCharts-R1 collects real-world charts and generates code, using the code as prior knowledge to produce highly accurate QA pairs. ChartMaster leverages real-world charts as seeds to stimulate diverse code outputs, employing multiple prompting strategies to maximize both instruction and replotted chart diversity.
>
> **Reward Design:**
> BigCharts-R1 introduces a text-based Chart Error Rate Reward aimed at improving QA accuracy, while ChartMaster employs multimodal reward functions to enhance chart-to-code generation. Our attribute reward encourages the model to capture key chart content such as text, and our visual reward ensures fine-grained visual alignment, enabling generated charts to closely match the originals in appearance—an aspect often overlooked by text-based rewards. Our approach has two main advantages: first, it effectively captures and preserves fine-grained visual details, as shown in the qualitative analysis in Figures 4 and 8; second, our ablation study (Appendix E.2) demonstrates that our method not only boosts chart-to-code generation performance but also improves QA accuracy, as enhanced chart-to-code capabilities lead to finer semantic understanding of charts, which transfers effectively to reasoning tasks.
>
>
> **Q8** : Calculate the visual reward with chart-aware visual encoders.
>
> | ChartMaster-7B |  \| Exec. Rate | \| Low-Level |||| | \| High-Level |
> |---------------|------------|-------|--------|-------|-------|-------|------------|
> |                   | \|          | \| Text  | Layout | Type  | Color | Avg.  | \| GPT-4o     |
> | w/ ChartCoder-Img. Enc.  | \|94.8       | \|83.1            | 93.1              | 71.1            | 65.7             | 78.2            | \|84.6               |
> | w/ ResNet-18             | \|93.8       | \|79.8            | 91.3              | 72.2            | 69.7             | 78.2            | \|85.1               |
>
> **A8** :  Thank you for your comment. In Table 5, we have compared the performance of different CNN-based visual encoders and concluded that deep visual features are effective. To further investigate, we use the chart-aware visual encoder from ChartCoder-7B to extract features. As shown in the table, the chart-aware encoder slightly improves execution rate and achieves better text and layout metrics, reflecting enhanced chart-specific feature extraction. However, it lags behind ResNet-18 in Type and Color metrics, suggesting that traditional CNNs may better capture texture and color information. Overall, both encoders show comparable average performance. We appreciate this suggestion and will continue exploring more specialized visual encoders in future work.
>
> The results are added to Appendix E.6. Thank you!

---

### Official Review · Reviewer_DZAz · 2025-11-01

**Soundness:** 3
**Presentation:** 3
**Contribution:** 3
**Rating:** 4
**Confidence:** 3

**Summary:**

This paper presents ChartMaster, a state-of-the-art chart-to-code model trained using a two-stage pipeline: supervised finetuning (SFT) and GRPO. To achieve this, the authors create the ReChart Prompt dataset which is a visually diverse chart-to-code dataset that generates synthetic chart-code pairs by “replotting” real-world images sourced from arXiv. For GRPO, the authors propose two novel reward metrics: visual attribute similarity (overlap in the layout/color attributes between the generated and ground truth chart/code) and visual similarity (measured using the cosine similarity between the geneated and reference chart using Resnet18 for feature extraction). The authors evaluate their model, ChartMaster, on three downstream tasks such as ChartMimic, ChartX, and Plot2Code, and the model achieves SOTA performance on most of them. Finally, the authors conducted a set of ablation studies to justify their design choices and done some qualitative analysis to showcase the superiority of their approach and model.

**Strengths:**

* The ReChart Prompt pipeline is quite novel and helps increase the visual diversity of the resulting dataset. It proposes a “chart replotting” technique that conditions the generation of synthetic chart-code pairs on real-world charts.


* The authors designed two novel reward functions for the GRPO algorithm with detailed ablations to justify and support their design choices (Tables  3, 4, and 5). Also, the resulting model ChartMaster achieves SOTA results on a variety of chart-to-code tasks.


* I believe the ReChart Prompt dataset could be valuable to the research community and could be extended to other domains such as QA and Fact Checking.

**Weaknesses:**

* The chart images are sourced from one source, arXiv, which may limit the visual and topics diversity in the dataset.

* The authors claim that their approach of replotting real-world chart images increases the diversity of the dataset compared to existing approaches that just prompts LLM to generate chart-code pairs. While I believe this is likely true, there’s no analysis to support this claim.

* The paper is only limited to chart-to-code which is a very niche task and doesn’t explore the potential of the proposed approach on more diverse chart understanding tasks such as QA and Fact checking.

**Questions:**

See weaknesses above.

---

> ### Author Response · Authors · 2025-11-23
>
> **Q1** :  Strengths in visual and topic diversity.
>
> **A1** :  Thank you for your insightful comment. We address your concerns from two perspectives:
>
> **1. arXiv source and topic diversity:**
> We select arXiv as our primary data source because it is a high-quality, open academic repository with clear chart images suitable for code generation. While our collected papers mainly come from the AI field, we do not restrict specific research topics. As a result, our dataset covers areas such as computer vision, natural language processing, robotics, and also includes charts from AI applications in physics, medicine, and biology. This topic diversity is reflected in Figure 3, where our dataset shows a significantly higher number of unique text attributes compared to existing datasets. To further enhance diversity, we plan to incorporate charts from additional sources and domains such as social sciences in future work.
>
>
> **2. Visual diversity:**
> The ReChartPrompt dataset demonstrates high visual diversity. As shown in Figure 3, 5 (Appendix), compared with existing Chart2Code datasets, ReChartPrompt contains a wider range of table contents and layout styles.
>
>
> **Q2** :  Evidence for visual and topic diversity.
>
> **A2** :  Thank you for your valuable feedback. Our approach leverages the broad distribution of real-world charts to prompt MLLMs, encouraging the generation of diverse code outputs that reflect authentic attribute and layout variations. In contrast, existing dataset relies on LLM prompts, often resulting in repetitive attribute patterns and less visual diversity.
>
> To support our claim, we conduct both quantitative and qualitative comparisons. As shown in Figure 3, we measure attribute diversity by counting unique values for key chart properties, and compare model performance after fine-tuning. Our ReChartPrompt dataset shows substantially higher diversity across all chart attributes and achieves better results. Figure 5 further illustrates the broader range of visual styles in our dataset. We hope these analyses clearly support our claim and address your concern.
>
>
> **Q3** :  Extension to chart understanding tasks.
>
> | Tiny Chart | ReChart Prompt | SFT  | RL   | \|ChartQA | \|ChartQAPro ||||||
> |----|------|--|---|-|-| -|-|-|-|-|
> |            |                |      |      | \|        | \|Factoid | Conversational | Hypothetical | Fact Checking | Multi Choice | Overall |
> | $\checkmark$ |              | $\checkmark$ |      | \|87.8    | \|26.7    | 39.7           | 41.7         | 38.5          | 35.5         | 34.2    |
> | $\checkmark$ | $\checkmark$  | $\checkmark$ |      | \|89.2    | \|27.5    | 42.1           | 36.0         | 45.0          | 39.2         | 37.9    |
> | $\checkmark$ | $\checkmark$  | $\checkmark$ | $\checkmark$ | \|89.8 | \|29.3    | 43.4           | 38.9         | 47.1          | 36.9         | 39.1    |
>
> **A3** :  Thank you for your valuable suggestion. Following your advice, we conduct further experiments on ChartQA and ChartQAPro benchmarks.
>
> Following ChartCoder, we incorporate the TinyChart dataset throughout the training process. Specifically, we first use 240K TinyChart instances for SFT on Qwen2.5-VL-7B as the baseline. Then, we jointly train the model with our own dataset during SFT. During GRPO, we use 24K QA samples from ChartQA, PlotQA, and DVQA subsets, applying an accuracy-based reward for QA and attribute/visual rewards for chart-to-code. Losses for QA and chart-to-code tasks are computed separately and averaged; other hyperparameters remain unchanged.
>
> As shown in table, incorporating ReChartPrompt data during SFT notably improves QA accuracy, especially for Fact Checking, with further gains from RL. This demonstrates that chart-to-code learning enhances the model’s fine-grained understanding of chart semantics and transfers effectively to reasoning tasks, resulting in better QA performance. These results are added to Appendix E.2.

---

### Official Review · Reviewer_Gapn · 2025-11-01

**Soundness:** 3
**Presentation:** 3
**Contribution:** 2
**Rating:** 4
**Confidence:** 4

**Summary:**

This paper addresses the task of chart-to-code generation, which aims to translate chart images into executable plotting code. It propose a complete data synthesis and distillation pipeline that constructs a large-scale, high-quality dataset (ReChartPrompt) from real-world scientific figures. Building upon this, the paper designs a reinforcement learning framework based on GRPO, incorporating a dual reward signal that jointly measures visual similarity and attribute-level semantic consistency between generated and reference charts. Finally, it trains a 7B-parameter model named ChartMaster, which achieves performance comparable to GPT-4o.

**Strengths:**

Originality:
While the overall training paradigm (data distillation → SFT → GRPO) follows established LLM fine-tuning practices, the paper demonstrates an original and well-motivated application of this pipeline to the underexplored domain of chart-to-code generation. The proposed ChartSimRL introduces a novel dual-reward design that jointly leverages visual and attribute similarity signals—an inventive adaptation of multimodal reward shaping to code generation tasks. This represents a meaningful step toward aligning visual and semantic fidelity in multimodal reasoning.
Quality:
The paper is technically solid and empirically thorough. The proposed components are well implemented and supported by extensive experiments across multiple benchmarks (ChartMimic, Plot2Code, ChartX). Ablation studies clearly isolate the effects of the dataset and each reward component, and results are reproducible with open-source models and frameworks. The training and evaluation pipeline is carefully described, suggesting high implementation quality.
Clarity:
The paper is well structured and clearly written. The motivation, methodology, and experimental design are logically presented with sufficient mathematical and procedural detail. Figures and tables are effectively used to illustrate qualitative and quantitative results, making the narrative easy to follow for both machine learning and vision-language audiences.
Significance:
The work makes a strong empirical contribution by achieving performance comparable to GPT-4o with an open-source 7B model, thus narrowing the gap between proprietary and community models in chart-to-code generation. The newly constructed ReChartPrompt dataset also provides lasting value to the research community as a more realistic and diverse resource for training multimodal reasoning systems. Together, these contributions make the paper a valuable and impactful addition to the field.

**Weaknesses:**

1. Insufficient analysis of the SFT–RL interplay.
The paper does not clearly isolate the contribution of the SFT and GRPO stages in the final model’s performance. Specifically, it remains unclear whether the improvement of ChartMaster over the SFT baseline arises from the GRPO phase itself or from the preceding supervised fine-tuning on ReChartPrompt. The authors did not conduct an experiment where Qwen2.5-VL-7B is directly fine-tuned with GRPO without SFT, which would have provided stronger evidence for the necessity of the full training pipeline. Without this, it is difficult to assess whether GRPO alone suffices for the chart-to-code domain or if SFT is an essential prerequisite to preserve general multimodal reasoning ability.
2. Lack of fine-grained quantitative analysis.
Although qualitative examples are informative, the evaluation does not fully leverage the fine-grained metrics available in ChartMimic, such as text accuracy, layout fidelity, chart type classification, and color consistency. A more detailed quantitative breakdown would help clarify which aspects of visual-semantic consistency are most improved by ChartSimRL. The current presentation may leave readers uncertain about where the model’s strengths and weaknesses lie within specific visual attributes.
3. Minor issues in limited methodological originality.
Although the paper proposes a well-executed adaptation of GRPO to the chart-to-code domain, the overall training pipeline closely follows the now-standard Distillation → SFT → GRPO paradigm used in recent works such as DeepSeek-R1 and Vision-R1. The novelty lies mainly in the domain-specific reward formulation (visual + attribute similarity), which is a practical but incremental design rather than a conceptual advance in RL or multimodal alignment. The authors could strengthen this aspect by providing deeper analysis or theoretical motivation for their reward shaping strategy, or by showing that their formulation generalizes to other multimodal generation tasks beyond chart-to-code.

**Questions:**

1. On the necessity of SFT before GRPO.
Could the authors clarify whether the supervised SFT stage is essential for achieving the reported performance of ChartMaster? Such an experiment would help disentangle how much of the performance gain originates from the reinforcement learning phase versus from the prior data distillation and SFT stages. Understanding this would also shed light on whether GRPO alone suffices for adapting pretrained multimodal models to the chart-to-code domain, or if SFT provides necessary grounding of fundamental reasoning capabilities.
2. On the relationship between reward components and fine-grained metrics.
The proposed dual reward combines attribute-level similarity (R_attr) and visual similarity (R_vis). Could the authors provide more evidence or analysis on how these two components correlate with fine-grained evaluation metrics—such as text accuracy, layout alignment, color consistency, or chart type fidelity—available in benchmarks like ChartMimic? Clarifying which reward component most influences each aspect of chart reconstruction quality would help readers better understand the strengths and limitations of the ChartSimRL design.
3. On the dependence of model performance on the teacher used in data distillation.
Given that the ReChartPrompt-240K dataset was generated using Qwen2.5-VL-72B as the teacher model, how sensitive is the final performance of ChartMaster to the teacher’s quality? In other words, if the same pipeline were applied with a stronger teacher model (e.g., GPT-4o) for data distillation, would the resulting student potentially exceed GPT-4o’s performance on chart-to-code benchmarks, or does the distillation process inherently cap the achievable quality? A discussion or small-scale experiment could clarify this dependency and help position the contribution relative to future stronger teachers.

---

> ### Author Response · Authors · 2025-11-23
> **Part 1/2**
>
> **Q1**: Disentangling SFT and GRPO contributions and assessing the necessity of SFT.
>
> | ReChartPrompt | ChartSimRL | \| ChartMimic      |      |       | \|  Plot2Code    |      |  | \| ChartX    |
> |-|-|-|-|-|-|-|-|-|
> |               |            |    \| Exec.Rate        | Low-Level       | High-Level       | \| Pass Rate     | Text-Match    | Rating    | \| GPT-score |
> |               |            | \|65.5             | 39.9            | 40.7             | \| 67.4          |  43.8          | 4.60      | \| 2.18      |
> | $\checkmark$             |            | \|91.1             | 73.7            | 80.9             | \| 80.3          | 59.3          | 5.34      | \| 2.36      |
> |               | $\checkmark$          | \|83.6             | 58.6            | 57.6             | \| 72.7          | 50.8          | 5.19      | \| 2.23      |
> | $\checkmark$             | $\checkmark$          | \|93.8             | 78.2            | 85.1             | \| 88.2          | 62.6          | 5.65      | \| 2.46      |
>
>
> **A1**:  Thank you for your feedback. As shown in the table, using GRPO alone does improve the model's performance compared to the baseline. However, the combination of SFT and GRPO achieves the best results. This is because the SFT stage enhances the model's multimodal reasoning abilities and helps it acquire chart-specific knowledge. This “cold start” phase allows the model to generate high-quality candidate outputs during the subsequent GRPO stage, further improving its capabilities in chart-to-code generation and achieving optimal performance.
>
> We have added this experiment and conclusion to Table 2 in the revised manuscript. Thank you.
>
> **Q2**: Fine-grained results and their relationship with $R_i^{\mathrm{attr}}$ and $R_i^{\mathrm{vis}}$.
> | ReChartPrompt | ChartSimRL | | \| Exec. Rate | \| Low-Level |||| | \| High-Level |
> |---------------|-----------------------|----------------------|------------|-------|--------|-------|-------|-------|------------|
> |               | $R_i^{\mathrm{attr}}$ | $R_i^{\mathrm{vis}}$ |   \|          | \| Text  | Layout | Type  | Color | Avg.  | \| GPT-4o     |
> |               |                       |                      | \| 65.5       | \| 35.2  | 58.1   | 37.8  | 28.3  | 39.9  | \| 40.7       |
> | $\checkmark$  |                       |                      | \| 91.1       | \| 75.6  | 87.8   | 67.0  | 64.3  | 73.7  | \| 80.9       |
> | $\checkmark$  | $\checkmark$           |                      | \| 92.1       | \| 80.1  | 90.2   | 69.5  | 65.1  | 76.2  | \| 83.9       |
> | $\checkmark$  |                       | $\checkmark$          | \| 92.1       | \| 79.8  | 90.6   | 71.8  | 68.7  | 77.7  | \| 84.3       |
> | $\checkmark$  | $\checkmark$           | $\checkmark$          | \| 93.8       | \| 79.8  | 91.3   | 72.2  | 69.7  | 78.2  | \| 85.1       |
>
> **A2**: Thank you for the insightful questions!
> As shown in the above table, SFT on ReChartPrompt significantly boosts all metrics. Adding either the attribute reward $R_i^{\mathrm{attr}}$ or visual reward $R_i^{\mathrm{attr}}$ further improves low-level metrics, but their effects differ.
>
> Specifically, $R_i^{\mathrm{attr}}$ mainly enhances text accuracy and layout fidelity, but has limited impact on color consistency. This is because $R_i^{\mathrm{attr}}$ relies on discrete matching, where both subtle and large color differences are treated as mismatches, even though larger discrepancies should be penalized more heavily. In contrast, the visual similarity reward ($R_i^{\mathrm{vis}}$), which evaluates global image features in a continuous manner, better captures approximate color and gradient variations, resulting in stronger gains in color consistency.
>
> Therefore, the optimal approach is to combine both reward mechanisms, leveraging their complementary strengths to achieve robust and fine-grained chart-to-code reconstruction.
>
> We have added this experiment to Appendix E.1 in the revised manuscript. Thank you.

---

> ### Author Response · Authors · 2025-11-23
> **Part 2/2**
>
> **Q3**: Deeper analysis and generalization.
>
> | Tiny Chart | ReChartPrompt | SFT  | RL   | \|ChartQA | \|ChartQAPro ||||||
> |----|------|--|---|-|-| -|-|-|-|-|
> |            |                |      |      | \|        | \|Factoid | Conversational | Hypothetical | Fact Checking | Multi Choice | Overall |
> | $\checkmark$ |              | $\checkmark$ |      | \|87.8    | \|26.7    | 39.7           | 41.7         | 38.5          | 35.5         | 34.2    |
> | $\checkmark$ | $\checkmark$  | $\checkmark$ |      | \|89.2    | \|27.5    | 42.1           | 36.0         | 45.0          | 39.2         | 37.9    |
> | $\checkmark$ | $\checkmark$  | $\checkmark$ | $\checkmark$ | \|89.8 | \|29.3    | 43.4           | 38.9         | 47.1          | 36.9         | 39.1    |
>
>
> **A3**: Thank you for your thoughtful suggestions. In A2, we provide a fine-grained quantitative analysis showing how each reward component correlates with low-level chart-to-code metrics. Our experiments demonstrate that the attribute reward mainly improves text accuracy and layout fidelity, while the visual reward is crucial for enhancing color consistency. Combined with extensive analyses in Tables 3, 4, and 5, our work offers a comprehensive evaluation of the reward shaping strategy.
>
> We further evaluate the generalization of our approach to QA tasks. Following ChartCoder, we incorporate the TinyChart dataset throughout the training process. Specifically, we first use 240K TinyChart instances for SFT on Qwen2.5-VL-7B as the baseline. Then, we jointly train the model with our own dataset during SFT. During GRPO, we use 24K QA samples from ChartQA, PlotQA, and DVQA subsets, applying an accuracy-based reward for QA and attribute/visual rewards for chart-to-code. Losses for QA and chart-to-code tasks are computed separately and averaged; other hyperparameters remain unchanged.
>
> The table above shows that incorporating ReChartPrompt data during SFT significantly improves QA accuracy, with further gains from RL. This is because chart-to-code learning enhances the model’s fine-grained understanding of chart semantics, enabling effective transfer to reasoning tasks. These results are added in Appendix E.2.
>
>
> **Q4**: Impact of teacher model.
>
>
> |               Method                             |   SFT | GRPO     |   \|ChartMimic      ||        |   \|Plot2Code         |  |    |  \|ChartX  |
> |-------------------------------------------|-----|------|-----------|-----------|------------|-----------|------------|--------|-----------|
> |                                           |     |      | \|Exec.Rate | Low-Level | High-Level | \|Pass Rate | Text-Match | Rating | \|GPT-score |
> | Qwen2.5-VL-7B                             |     |      |   \|65.5    |   39.9    |   40.7     |   \|67.4    |   43.8     | 4.60   | \|2.18      |
> | Qwen2.5-VL-72B                            |     |      |   \|88.5    |   72.7    |   79.1     |  \|84.8    |   68.4     | 6.83   | \|2.52      |
> | Qwen3-VL-235B-A22B-Instruct               |     |      |   \|94.0    |   79.1    |   82.3     |   \|90.1    |   56.3     | 6.49   | \|2.94      |
> | Use Qwen2.5-VL-72B as Teacher Model       | $\checkmark$   |      |   \|91.1    |   73.7    |   80.9     |   \|80.3    |   59.3     | 5.34   | \|2.36      |
> |                                           | $\checkmark$   | $\checkmark$    |   \|93.8    |   78.2    |   85.1     |   \|88.2    |   62.6     | 5.65   | \|2.46      |
> | UseQwen3-VL-235B as Teacher Model         | $\checkmark$   |      |   \|91.1    |   75.3    |   81.5     |   \|82.5    |   64.3     | 5.57   | \|2.48      |
> |                                           | $\checkmark$   | $\checkmark$    |   \|95.1    |   79.4    |   86.2     |   \|88.6    |   65.7     | 5.93   | \|2.53      |
>
>
> **A4**: Thank you for your insightful question. We select Qwen3-VL-235B-A22B-Instruct as the stronger teacher instead of GPT-4o because it is open-source and highly capable, facilitating reproducibility. To ensure a fair comparison, we generate a new 240K chart-to-code dataset using the Qwen3-VL-235B-A22B-Instruct model, while keeping all other data construction and training settings unchanged.
>
> As shown in table, ChartMaster's performance improves significantly when a stronger teacher model is used for data distillation, demonstrating that teacher quality substantially impacts student performance. Importantly, our method enables the student model to closely match and even outperform the teacher on certain metrics, evidencing the effectiveness of our approach in leveraging high-quality teacher knowledge.
>
> We have added this experiment to Appendix E.3 in the revised manuscript. Thank you.

---

> > ### Comment · Reviewer_Gapn · 2025-11-28
> >
> > After reading the response, I'd like to raise the score to 6, if possible.

---

### Author Response · Authors · 2025-12-01
**Summary for Area Chair – Part 1/2**

We thank the AC and all reviewers for their time and valuable feedback. We understand the current situation and deeply appreciate the AC’s efforts. This summary highlights our core contributions and how we have addressed the main concerns raised during the review.

---

## Paper Snapshot

**Task:** *Chart-to-Code Generation.*  Given a chart image, the model generates executable code to reproduce the chart.

**Challenges:**

1. *Lack of diversity in existing datasets.*   Existing datasets use predefined seed attributes for LLM-based code generation, causing repetitive outputs and limited chart diversity (see Fig.~5).

2. *SFT Limitations.*  Next-token prediction does not ensure visual consistency between generated and reference charts.

**Solutions:**

1. *ReChartPrompt:*  A pipeline that uses real-world charts to replace seed attributes and enhance diversity. Charts are sourced from arXiv, and code generation with an open-source model plus automated filtering yields a free, high-quality dataset without manual annotation.

2. *ChartSimRL:*  A GRPO-based algorithm that combines attribute and visual rewards. Each candidate chart is scored by attribute set overlap and visual similarity with the original chart, guiding the model to achieve better visual consistency.

**Main Results:**

|               Model                             |    \|ChartMimic      ||        |   \|Plot2Code         |  |    |  \|ChartX  |
|----|-----|-----|---|-----------|------------|--------|-----------|
|                                           |     \|Exec.Rate | Low-Level | High-Level | \|Pass Rate | Text-Match | Rating | \|GPT-score |
| GPT-4o                         |  \|93.2      | 79.0      | 83.5       |  \|88.6      | 56.3       | 5.71   |  \|2.36      |
| Qwen2.5-VL-7B (Base.)          |  \|65.5      | 39.9      | 40.7       |  \|67.4      | 43.8       | 4.60   |  \|2.18      |
| Base. + ReChartPrompt          |  \|91.1      | 73.7      | 80.9       |  \|80.3      | 59.3       | 5.34   |  \|2.36      |
| Base. + ReChartPrompt + ChartSimRL |  \|93.8  | 78.2      | 85.1       |  \|88.2      | 62.6       | 5.65   |  \|2.46      |
| Comparison with GPT-4o         |  \|**+0.6**  | **-0.8**  | **+1.6**   |  \|**-0.4**  | **+6.3**   | **-0.06** | \| **+0.10** |

**Contributions:**

1. **ReChartPrompt-240K:** A large-scale and diverse chart-to-code generation dataset.
2. **ChartSimRL:** A novel algorithm for chart-to-code generation.
3. **ChartMaster:** A 7B model that achieves performance comparable to GPT-4o.
4. **Open-source release:** All datasets, code, and models will be fully released.

---

> ### Author Response · Authors · 2025-12-01
> **Summary for Area Chair – Part 2/2**
>
> ---
>
> ## Key Concerns and Response
>
> For convenience, we refer to the four reviewers Gapn, DZAz, 5icT, and ER2g as R1, R2, R3, and R4.
>
> All reviewers consider our method solid and agree that its soundness is good. The dataset is highly valuable (R1, R2, R4), and the reward mechanism is both novel (R1, R2) and reasonable (R3, R4). Our experiments are comprehensive (R1, R4), and the model’s strong performance is highly praised (R1, R2, R3, R4). We are greatly encouraged by the recognition of the novelty and contributions of our work.
>
> However, reviewers also raised two main concerns:
> 1. Additional details are needed regarding the data processing and experiments, such as code-render success rate and results of baseline models trained directly with ChartSimRL.
> 2. Further exploratory experiments could help to uncover more potential of ChartMaster.
>
>
>
> | **Key Concerns** | **Response** |
> |------------------|--------------|
> | Lack of detailed data audit and filtering validation (R3, R4) | We provide further audit details in Appendix A, including image counts at each stage and code-render success rates. Supplementary Figure A provides manual inspection results, confirming that noisy samples are effectively filtered, while Figure B verifies no data leakage between train and test sets via image hashing. Overall, ReChartPrompt is a high-quality, diverse dataset, as evidenced by significant performance gains in the main results table. |
> | Lack of standalone ChartSimRL and detailed low-level metric results (R1) | Table 2 shows strong performance when training with ChartSimRL alone. However, SFT is essential for foundational knowledge and an effective cold start, so combining SFT with ChartSimRL yields optimal results. Detailed low-level metrics in Appendix E.1 confirm the complementary effects of the dual rewards: attribute reward excels in text, while visual reward is key for color consistency. |
> | Exploring our method’s generalization to additional tasks and more challenging samples (R1, R2, R3) | Appendix E.2 shows that our method improves semantic and visual extraction, leading to better performance on ChartQA and other VQA tasks. Supplementary Figure C shows that ChartMaster performs robustly on complex charts in stress tests. |
> | Exploring the impact of stronger teacher models (R1) | Appendix E.3 shows that student model performance improves with stronger teachers, and our method enables the student to surpass the stronger teacher on specific benchmarks. |
> | Exploring the impact of different visual encoders and potential overfitting in visual reward (R3) | In Appendix E.6, we replace ResNet-18 with a chart-aware visual encoder, which improves text and layout scores but reduces color accuracy, confirming ResNet-18 as an effective choice. Visualization experiments in Supplementary Figure 10 confirm that the visual reward does not overfit to style surrogates. |
>
> ---
>
> ## Conclusion
>
> We truly appreciate the thoughtful feedback from all reviewers. We’re glad our method, experiments, and model performance were recognized as solid and comprehensive. We have carefully addressed all concerns with more details and additional exploratory experiments, leading to a much improved and more rigorous manuscript.
>
> Thank you again to the AC and reviewers for your time and consideration. We hope our efforts will be helpful to everyone.

---

### Meta-Review · Area_Chair_uvwL · 2026-01-05

**Summary:**

This paper introduces a framework for chart-to-code generation utilizing a large-scale, real-world dataset and a dual-reward reinforcement learning algorithm. The proposed framework achieves impressive empirical results. During the rebuttal, the authors provided significant technical rigor by conducting an extensive Image Hashing Audit to effectively rule out data leakage concerns and showcased positive transfer to ChartQA tasks. However, despite the high quality of execution, the core contribution is primarily viewed as a domain-specific engineering adaptation of standard distillation and reinforcement learning paradigms. Persistent bottlenecks in precise numerical extraction from complex charts and a heavy dependency on teacher model outputs ultimately limit the work's fundamental algorithmic innovation and its broader academic impact within the competitive scope of this conference.

**Reviewer Concerns:**

Addressed:

The authors resolved critical concerns regarding data leakage through a comprehensive image hashing audit of the entire dataset. They also provided evidence of the model's generalization capabilities by demonstrating improved performance on downstream tasks, such as ChartQA.

Outstanding:

The primary critique regarding incremental novelty remains unaddressed; the training pipeline follows established industry standards for model tuning without introducing transformative insights into multimodal reasoning. Furthermore, the lack of robustness in exact numerical extraction persists as a significant technical limitation for the task's primary application.

**Reviewer Scores:**

Reviewer Gapn: Actively engaged in the discussion and upgraded their score from 4 to 6, acknowledging the rigor of the data audit.

Reviewer ER2g: Maintained their initial assessment; the score reflects a consistent appreciation for the empirical results.

Reviewer DZAz: Did not provide additional feedback during the discussion phase; their original perspective on the work's scope remains as stated in the initial review.

Reviewer 5icT: Did not provide further comments during the discussion.

---

### Decision · Program_Chairs · 2026-01-26

Reject